# EnzyControl: Adding Functional and Substrate-Specific Control for Enzyme Backbone Generation

**Chao Song**[1][*], **Zhiyuan Liu**[2][*], **Han Huang**[3], **Liang Wang**[4],
**Qiong Wang**[1], **Jianyu Shi**[1], **Hui Yu**[1][†], **Yihang Zhou**[2][†], **Yang Zhang**[2][†]

[1]Northwestern Polytechnical University,   [2]National University of Singapore
[3]The Chinese University of Hong Kong,   [4]Institute of Automation at CAS

csong@mail.nwpu.edu.cn, zhiyuan@nus.edu.sg
huiyu@nwpu.edu.cn, yihangjoe@foxmail.com, zhang@nus.edu.sg

## Abstract

Designing enzyme backbones with substrate-specific functionality is a critical challenge in computational protein engineering. Current generative models excel in protein design but face limitations in binding data, substrate-specific control, and flexibility for de novo enzyme backbone generation. To address this, we introduce **EnzyBind**, a dataset with 11,100 experimentally validated enzyme-substrate pairs specifically curated from PDBbind. Building on this, we propose **EnzyControl**, a method that enables functional and substrate-specific control in enzyme backbone generation. Our approach generates enzyme backbones conditioned on MSA-annotated catalytic sites and their corresponding substrates, which are automatically extracted from curated enzyme-substrate data. At the core of EnzyControl is **EnzyAdapter**, a lightweight, modular component integrated into a pretrained motif-scaffolding model, allowing it to become substrate-aware. A two-stage training paradigm further refines the model's ability to generate accurate and functional enzyme structures. Experiments show that our EnzyControl achieves the best performance across structural and functional metrics on EnzyBind and EnzyBench benchmarks, with particularly notable improvements of 13% in designability and 13% in catalytic efficiency compared to the baseline models. The code is released at https://github.com/Vecteur-libre/EnzyControl.

## 1 Introduction

Enzymes are catalysts that drive essential chemical transformations with exceptional selectivity and effectiveness, enabling critical biological processes and industrial applications [1, 2]. Their engineerability allows precise optimization, making enzyme design a vital research area spanning pharmaceuticals [3, 4], specialty chemicals [5], biofuels [6], food production [7], and material synthesis [8]. In enzymatic reactions, binding specific small-molecule substrates is the key [9, 10], yet designing enzymes with precise substrate specificity remains a major challenge.

While enzyme design is often regarded as a subfield of protein design, the two tasks exhibit both similarities and fundamental differences. General protein design has made remarkable progress across several directions, including sequence generation guided by fitness landscapes [11, 12, 13, 14, 15, 16], structure-to-sequence generation based on predefined backbones [17, 18, 19, 20, 21, 22], and co-design of sequence and structure [23, 24, 25]. However, these approaches are not directly applicable

---

[*]Equal contribution.    † Corresponding authors.

to enzyme design, due to unique challenges specific to enzymes. Unlike general proteins, enzymes must satisfy strict substrate-specific binding requirements, exhibit evolutionarily conserved functional sites, and maintain catalytic conformations that are highly sensitive to subtle structural variations [26, 27]. Consequently, effective enzyme design must account for substrate interactions, functional site preservation, and conformational constraints, which are often neglected in protein design methods.

Several lines of work have emerged focusing specifically on enzyme design [4, 28, 29, 30, 31, 32]. Nonetheless, existing methods still face significant limitations.

- **Neglect of functional site conservation.** Previous protein generation methods either ignore functional sites or randomly select them during generation, resulting in poor catalytic function and high false positive rates. Although methods like EnzyGen [31] attempt to consider functional sites, they fail to ensure structural designability due to inadequate backbone generation quality.

- **Ignoring substrate molecules during generation.** Most previous works [33, 34, 35] design protein backbones without considering substrate interactions, limiting their applicability to real-world catalytic tasks. While EnzyGen uses the substrate to filter generated enzymes after generation, it does not incorporate substrate information during the generation process, making it unable to tailor enzyme structures for the substrate.

- **Absence of high-quality benchmarks.** Existing benchmarks [31, 36] are mostly synthetic with limited experimental grounding and lack evaluation protocols tailored to enzyme families. Since enzymes are classified not by structure but by the chemical reactions they catalyze (*i.e.,* enzyme commission (EC) number, App. C), their utility depends on how effectively they perform these reactions [37]. As a result, meaningful evaluation demands benchmarks designed around enzyme families and their functional roles.

To address the challenges of model design, we develop **EnzyControl**, a framework that extends standard motif-scaffolding models (*i.e.,* FrameFlow [38]) for substrate-aware enzyme backbone generation. EnzyControl consists of **three key components**: **(1)** A base network pretrained for motif-scaffolding. Specifically, we identify evolutionarily conserved functional motifs through multiple sequence alignments (MSA). These functional sites will be used to condition the base network, ensuring that key catalytic features are retained during generation. **(2)** The EnzyAdapter, a lightweight adapter that injects substrate information into the base network. It employs a cross-modal projector [39, 40, 41, 42] to bridge the modality gap between substrate and enzyme, and uses cross-attention [43, 44] layers to condition the generation on substrate without altering the base network. **(3)** EnzyControl includes a two-stage training strategy to facilitate stable and efficient learning. In the first stage, only the EnzyAdapter are trained to align substrate features with enzyme structures, preserving the pretrained parameters. In the second stage, the full model is fine-tuned using a Low-Rank Adaptation (LoRA) approach [45, 46, 47], with continued updates to the adapter guided by the generation loss. Our approach effectively integrates functional site conservation and substrate-aware conditioning, leading to higher fidelity in enzyme backbone generation.

Resolving the absence of high-quality benchmarks, we construct **EnzyBind**, a curated dataset of 11,100 enzyme-substrate pairs derived from PDBbind [48]. Each entry is enriched with functional site annotations via MSA. Further, we leverage enzyme family classification for evaluating the consistency of enzyme commission (EC) number between the generated sample and its target native enzyme, thereby providing a more rigorous evaluation framework.

We benckmark EnzyControl on EnzyBind, evaluating the generated enzyme backbones across multiple structural and functional metrics. Experiments show that EnzyControl achieves 0.7160 in designability, a significant **13%** relative improvement compared to the second-best model (see Table 1). It also demonstrates significantly improved catalytic efficiency ($k$cat) and functional alignment (EC match rate), achieving **13%** and **10%** improvements, respectively, over the suboptimal baselines. EnzyControl also achieves **3%** improvement of binding affinity than the second-best model on EnzyBench (Table 7). Additional quantitative analyses further highlight its strong residue efficiency (Fig. 9). In particular, EnzyControl consistently generates sequences that are approximately **30%** shorter, while maintaining comparable $k$cat values across all catalytic efficiency ranges—indicating its ability to produce compact, functionally robust designs suitable for practical applications.

## 2    Related Work

**Enzyme Design Applications.**    Designing effective enzymes remains a core challenge, particularly in identifying active sites and optimizing functional properties. Common strategies fall into three main categories: semi-rational design [49, 50, 51], rational design [52, 53, 54], and de novo design [55]. Semi-rational design leverages known structures and experimental data to guide site-directed mutagenesis near the catalytic site [56, 57], with residues selected based on structural insights and prior knowledge [58]. Rational design relies more heavily on computational modeling [59], using tools such as molecular dynamics and quantum mechanical simulations to explore enzyme–substrate interactions and reaction mechanisms [60, 61]. Both approaches aim to improve or repurpose natural enzymes. In contrast, de novo design constructs entirely new enzymes by embedding catalytic motifs into synthetic scaffolds, often simplifying structure to focus on function [32]. EnzyControl bridges rational and de novo design with a modular adapter for substrate-aware enzyme backbone generation, and leverages MSA to improve the efficiency of active site annotation.

**Methods for Motif-Scaffolding Task.**    The task is to create proteins with functional properties conferred through a prespecified arrangement of residues known as a motif. The problem is to design the remainder of the protein, called the scaffold, that harbors the motif. Early approaches to the motif-scaffolding problem primarily relied on assembling scaffolds from pre-defined protein fragments. These methods are limited by their dependence on finding suitable matches within the Protein Data Bank (PDB) and often struggle to accommodate slight structural mismatches between the scaffold and the motif [62, 63, 64, 65]. More recent models, such as RFdiffusion [66] and FrameFlow [38], represent a shift toward generative modeling. These approaches use diffusion or flow matching models [67, 68, 69] conditioned on the motif's structure and/or sequence, while generating only the surrounding scaffold. However, they cannot incorporate additional design constraints such as substrate specificity. Our method addresses this limitation by enabling scaffold generation conditioned on a broader range of inputs, expanding the applicability of motif-scaffolding.

## 3    EnzyBind: High-Quality Enzyme Dataset

A significant limitation of existing datasets, such as EnzymeMap [70] and ReactZyme [71], is the absence of precise pocket information. These datasets only provide protein sequences and SMILES representations, which are primarily used for predicting EC numbers. Although EnzymeFill [72] addresses this issue by introducing a synthetic dataset with precise pocket structures and substrate conformations, its reliability is hindered by the lack of experimental validation in wet-lab settings. To overcome this limitation, we present EnzyBind, a novel dataset that includes precise pocket structures with substrate conformations, experimentally validated in wet-lab environments. EnzyBind is specifically designed to support enzyme catalytic backbone generation tasks.

**Data Source.**    To construct the EnzyBind dataset, we curated enzyme-substrate complexes from the PDBBind database (Fig. 1). Complexes that could not be processed using the RDKit library [73] were excluded. We then cleaned all remaining PDB files following a standardized procedure (detailed in App. D). This preprocessing pipeline resulted in a final dataset of approximately 11,000 enzyme-substrate complexes, covering six fundamental catalytic types, as illustrated in Fig. 2.

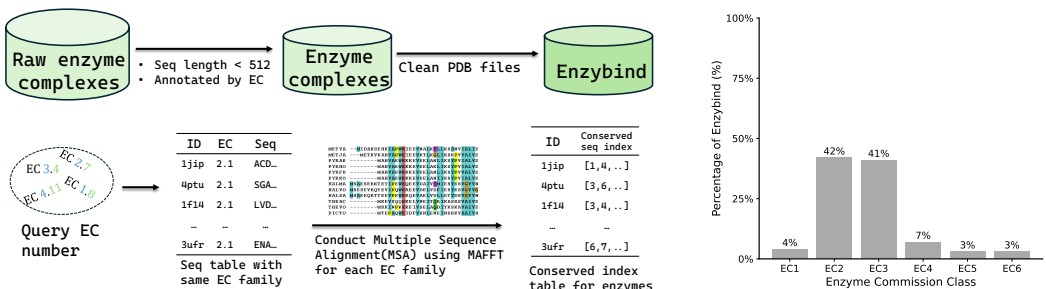

Figure 1: Dataset collection and preprocessing.          Figure 2: EC distribution.

**Functional Sites Extraction.**    To guide enzyme backbone generation toward functional outcomes, we first identified functional sites typically associated with conserved sequences. Building on prior

work [31], we used MSA to uncover evolutionarily conserved regions across enzymes belonging to the same second-level EC number, using the MAFFT software [74].

# 4 Proposed Method: EnzyControl

In this section, we present EnzyControl, a framework that can extend the motif-scaffolding model for substrate-specific enzyme backbone generation. We begin by introducing flow matching. Then, we present EnzyControl's key components: (1) a base network pretrained for motif-scaffolding; (2) an EnzyAdapter that adapts the base network for substrate-specific enzyme generation; and (3) a two-stage training strategy effectively combining the base network and EnzyAdapter.

## 4.1 Preliminary: Flow Matching

Flow matching (FM) [75, 76, 34, 77] is a generative modeling technique inspired by the strengths of diffusion models, offering a more efficient and stable sampling process. Given access to empirical observations of data distribution $\mathbf{x}_1 \sim p_1$ and noise distribution $\mathbf{x}_0 \sim p_0$, the goal of FM is to estimate a coupling $\pi(p_0, p_1)$ that describes the evolution between the two distributions. This objective could be formulated as solving an oridinart differential equation (ODE): $d\mathbf{x}_t = v(\mathbf{x}_t, t)dt$, on time $t \in [0, 1]$, where the vector field $v : \mathbb{R}^d \times [0, 1] \to \mathbb{R}^d$ is set to drive the flow from $p_0$ to $p_1$ and $\mathbf{x}_t$ lies along the interpolation between $\mathbf{x}_0$ and $\mathbf{x}_1$ with time $t$. We can parameterize the drift (vector field) by $v_\theta(\mathbf{x}_t, t)$ with an expressive learner (a neural network, for example) and estimate $\theta$ by solving a simple least square regression problem:

$$\hat{\theta} = \arg\min_\theta \mathbb{E}_{t, \mathbf{x}_t} \left[ \|v(\mathbf{x}_t, t) - v_\theta(\mathbf{x}_t, t)\|_2^2 \right]. \tag{1}$$

With this estimation, we can do backward sampling by taking $\hat{\mathbf{x}}_1 = \int_0^1 v_\theta(\mathbf{x}_t, t)dt$ since we have access to the noise $\mathbf{x}_0 \sim p_0$, and solve the integration with numerical integrators [78, 79].

## 4.2 EnzyControl's Base Network

**Notations and Problem Formulation.** An enzyme is a chain of amino acids (residues) linked by peptide bonds, which folds into a specific 3D structure. We represent the backbone of an enzyme with $N$ residues as $\mathbf{T} = [T^{(1)}, ..., T^{(N)}]$. Each frame $T = (\mathbf{r}, \mathbf{x}) \in \mathrm{SE}(3)$ encodes the atom positions of a residue, where $\mathbf{r} \in \mathrm{SO}(3)$ is a rotation matrix and $\mathbf{x} \in \mathbb{R}^3$ is a translation vector (details can be found in App. E.1). Within an enzyme backbone, we define a subset of residues $\mathbf{M} = \{T^{(i_1)}, \ldots, T^{(i_k)}\}$ as *functional sites*, where $\{i_1, \ldots, i_k\} \subset \{1, \ldots, N\}$. The remaining residues (called *scaffold*) are denoted as $\mathbf{S}$, such that $\mathbf{T} = \mathbf{M} \cup \mathbf{S}$. We also represent the enzyme's substrate as a chemical graph $\mathcal{G}$. Given the functional sites $\mathbf{M}$ and the substrate $\mathcal{G}$, our goal is to generate a compatible enzyme backbone by sampling the scaffold $\mathbf{S}$ from the conditional distribution $p(\mathbf{S}|\mathbf{M}, \mathcal{G})$.

**Conditional Enzyme Generation.** Throughout this section, we use FrameFlow [38] as our base network for motif-scaffolding. We adapt FrameFlow to accept both functional sites $\mathbf{M}$ (*i.e.,* motif-scaffolding) [80, 81] and substrate information $\mathcal{G}$ as conditions, and further conduct enzyme backbone generation. Given the conditions, the goal is to generate an enzyme backbone that preserves the spatial arrangement of $\mathbf{M}$ and can fit $\mathcal{G}$. Formally, the model needs to predict the conditional vector field $v(\mathbf{S}_t, t|\mathbf{S}_1, \mathbf{M}, \mathcal{G})$. Assuming conditional independence, the vector field can be simplified to $v(\mathbf{S}_t, t|\mathbf{S}_1, \mathcal{G})$. As functional sites $\mathbf{M}$ provides crucial information about the target structure $\mathbf{S}_1$, the predicted vector field is set to $v_\theta(\mathbf{S}_t, t|\mathbf{M}, \mathcal{G})$. Following the SE(3) representation of enzyme backbones (SE(3) = SO(3) × $\mathbb{R}^3$ [82]), the training objective minimizes the squared distance between the ground truth and predicted vector fields:

$$\mathbb{E}\left[ \|\mathbf{v}_\mathbb{R}(\mathbf{x}_t, t|\mathbf{x}_1) - \hat{\mathbf{v}}_\mathbb{R}(\mathbf{S}_t, t|\mathbf{M}, \mathcal{G})\|_\mathbb{R}^2 + \|\mathbf{v}_{\mathrm{SO}(3)}(\mathbf{r}_t, t|\mathbf{r}_1) - \hat{\mathbf{v}}_{\mathrm{SO}(3)}(\mathbf{S}_t, t|\mathbf{M}, \mathcal{G})\|_{\mathrm{SO}(3)}^2 \right]. \tag{2}$$

Where $\hat{\mathbf{v}}_\mathbb{R}$ and $\hat{\mathbf{v}}_{\mathrm{SO}(3)}$ are predicted vector fields of transition vector and rotation matrix.

To support SE(3)-equivariant generation, enzyme structures are represented as 3D $k$-nearest-neighbor ($k$-NN) graphs over residue frames, enabling spatially-aware message passing. Specifically, each node corresponds to a residue and edges connect spatially neighboring residues (*cf.* Fig. 3). The initial node embeddings $\boldsymbol{h}_0 = [h_0^1, ..., h_0^N] \in \mathbb{R}^{N \times D_h}$ are computed based on residue indices and the

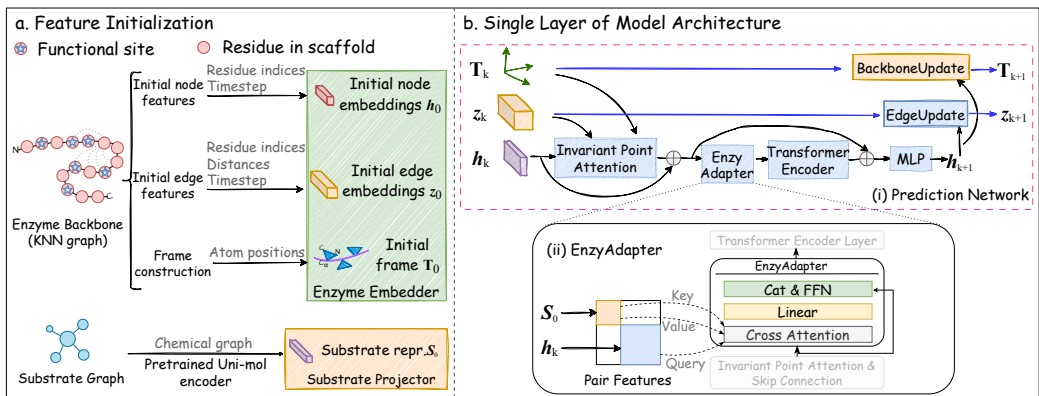

Figure 3: EnzyControl is a flexible approach for the conditional backbone generation of enzymes. (a) Feature Initialization involves obtaining initial node embeddings and edge embeddings, constructing initial frames for enzyme backbones, and initializing pretrained features for substrates. (b) Single-layer structure prediction network with EnzyAdapter.

current timestep. Edge embeddings $z_0 \in \mathbb{R}^{N \times N \times D_z}$ are initialized with residue indices, timestep, and relative sequence distances. They are further refined via self-conditioning using binned pairwise distance matrix between the model's $C_\alpha$ predictions. Each residue frame $\mathbf{T}_0^n = (\mathbf{r}, \mathbf{x}) \in \text{SE}(3)$ is initialized from backbone atoms, following AlphaFold2 (see App. E.1). These initialized embeddings are then processed Invariant Point Attention (IPA) [83] to capture spatial features in an SE(3)-equivariant manner. Transformer layers are interleaved between IPA layers to capture sequence-level dependencies. The model updates node embeddings using IPA and propagates these updates to edges through the EdgeUpdate module, which performs standard message passing. Finally, the BackboneUpdate module applies linear layers to predict translation and rotation updates to each frame. More method details are in App. E.3

## 4.3 EnzyAdapter and Two-Stage Training Strategy

We introduce EnzyAdapter, a modular architecture that enhances basic motif-scaffolding models with substrate awareness. EnzyAdapter is model-agnostic and modular, making it compatible with various backbone generation architectures.

**Substrate Feature Initialization.** In line with the design principles of AtomicFlow [84], we represent the substrate using its chemical graph rather than its 3D conformer, as the binding position of the substrate is typically unknown in advance. To get well-aligned substrate properties, we use a pretrained Uni-Mol [85] encoder model to extract substrate features. Uni-Mol is a flexible and scalable molecular pretraining model trained on 209 million molecular conformers, endowing it with rich knowledge of chemical structures and interactions. Given that our downstream dataset comprises only approximately 11,000 enzyme–substrate pairs, we freeze the Uni-Mol encoder during training to preserve the generalizable representations learned from large-scale pretraining and to avoid overfitting. To effectively decompose the substrate embedding, we use a small trainable projector. The projector we used in this study consists of two linear layers and a Layer Normalization [86]. This projector maps the Uni-Mol–derived substrate embeddings into the enzyme representation space, enabling the model to incorporate substrate-specific information in a flexible and task-aware manner without compromising the robustness of the pretrained encoder. This design strikes a principled balance between leveraging broad chemical knowledge and adapting to the specific demands of enzyme–substrate interaction modeling. Formally, given the input graph $\mathcal{G}$, the process is depicted as:

$$\boldsymbol{S}_0 = \text{Projector}(\text{UniMol}(\mathcal{G})), \quad \boldsymbol{S}_0 \in \mathbb{R}^{D_s}, \tag{3}$$

where $D_s$ denotes the embedding dimensions, and details of the projector can be found in App. E.2.

**EnzyAdapter Architecture.** In the original FrameFlow model, the enzyme features (nodes, edges and frames) are input directly into the IPA layers, without considering substrate interactions. A straightforward method to insert substrate features is to concatenate node features and substrate features and then feed them into the IPA layers [72]. However, we found this approach to be insufficiently effective. Instead, we propose EnzyAdapter where the used cross-attention layers for

substrate features and node features are separate, as depicted in Fig. 3(b). To be specific, we add EnzyAdapter for each layer to insert substrate features. Given the substrate features $\boldsymbol{S}_0$ and the node features in the $k$-th layer $\boldsymbol{h}_k$, the output of cross-attention $\boldsymbol{c}_k$ is computed as follows:

$$\boldsymbol{c}_k = \text{Attn}(\boldsymbol{Q}, \boldsymbol{K}, \boldsymbol{V}) = \text{Softmax}\left(\frac{\boldsymbol{Q}(\boldsymbol{K})^T}{\sqrt{d}}\boldsymbol{V}\right), \quad \boldsymbol{c}_k \in \mathbb{R}^{N \times D_h}, \tag{4}$$

where $\boldsymbol{Q} = \boldsymbol{h}_k\boldsymbol{W}_q$ is the query matrix from the node features, $\boldsymbol{K} = \boldsymbol{S}_0\boldsymbol{W}_k$ and $\boldsymbol{V} = \boldsymbol{S}_0\boldsymbol{W}_v$ are the key, and values matrices from the substrate features. $\boldsymbol{W}_q$, $\boldsymbol{W}_k$ and $\boldsymbol{W}_v$ are the corresponding weight matrices. Then, $\boldsymbol{c}_k$ is fed into a linear layer before concatenating it to the output of IPA. Hence, the final formulation of EnzyAdapter is defined as:

$$\boldsymbol{c}_k^{\text{new}} = \text{Linear}(\text{Concat}(\text{Linear}(\boldsymbol{c}_k), \boldsymbol{h}_k)), \quad \boldsymbol{c}_k^{\text{new}} \in \mathbb{R}^{D_h}. \tag{5}$$

**Two-stage Training Paradigm.** We adopt a two-stage training strategy to enhance both stability and efficiency (Fig. 4). In the first stage, we focus on learning substrate-specific features by training only the projector and EnzyAdapter, while keeping the other parts of the prediction network frozen. This allows the model to align substrates with their corresponding enzymes without interference from the backbone. In the second stage, we fine-tune the entire prediction network and the enzyme embedder using LoRA. Simultaneously, we continue updating the projector and EnzyAdapter, now guided by the generation loss. This joint optimization ensures that all components remain consistent and aligned with the overall objective.

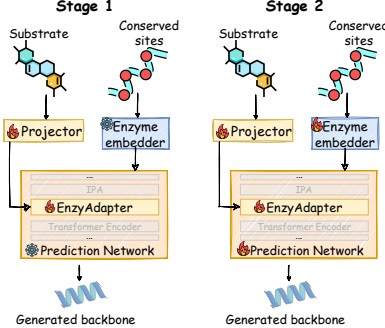

Figure 4: Two-stage training paradigm.

# 5 Experiments

## 5.1 Setup

**Implementation Details.** We use FrameFlow as the pretrained foundation model and fine-tune it on EnzyBind. To improve training efficiency, we group enzymes of the same sequence length into the same batch, minimizing padding overhead. For each input, we generate 20 backbone structures. ProteinMPNN [20] is then used to design 5 sequences per backbone, and each sequence's structure is predicted using ESMFold [87]. Additional implementation details are provided in App. F.1.

**Baseline Models.** We compare EnzyControl with EnzyGen, the current state-of-the-art enzyme generation model, which we retrain on EnzyBind for fair comparison. As models explicitly designed for enzyme backbone generation are limited, we also include several motif-scaffolding baselines: PROTSEED [24], RFDiffusion [66], Chroma [88], FADiff [81], RFDIffusionAA [89], Proteus [90] and Proteina [91]. RFDiffusion, RFDiffusionAA and Chroma are evaluated using their publicly released checkpoints, as training scripts are not available.

**Evaluation Metrics.** We evaluate the quality of generated enzyme backbones using both structural and functional metrics. To ensure a fair and consistent comparison across all methods, we adopt a unified evaluation pipeline. Specifically, for every method, the generated backbone structures are first processed through inverse folding using ProteinMPNN to obtain corresponding sequences. All-atom structures are then predicted from these sequences using ESMFold. All reported metrics are computed on these ESMFold-predicted structures, ensuring that differences in evaluation outcomes reflect genuine differences in backbone design rather than variations in sequence modeling or structure prediction protocols.

To assess structural consistency, we use two standard measures: TM-score (scTM), where higher values indicate closer alignment to the native structure, and $C_\alpha$-RMSD (scRMSD), where lower values are better. Following FoldFlow [77], we report the designability rate as the fraction of generated backbones with scRMSD < 2Å. Since Chroma defines designable backbones as those with scTM > 0.5, we also include the proportion meeting this threshold.

Because EnzyControl focuses on function-aware generation, we include additional metrics to evaluate functional relevance. First, we assess enzymatic function through EC number prediction using

Table 1: Evaluation of structural and functional validity of the generated enzyme backbones on EnzyBind. The best-performing results are marked in **bold**, and the second-best results are underlined.

| Model | Self Consistency | | Enzyme Property | | Substrate Specificity | | AAR | RMSD | Diversity | Novelty | Succ. Rate |
|---|---|---|---|---|---|---|---|---|---|---|---|
| | >0.5scTM | Design. | EC Match Rate | $k$cat | Bind. Aff. | ESP Score | | | | | |
| Eval Data | NA | NA | NA | NA | −8.6121 | NA | NA | NA | NA | NA | NA |
| EnzyGen | 0.0421 | 0.0263 | 0.4385 | 1.4142 | −6.7517 | 0.4724 | 0.0566 | 12.6075 | 0.1226 | 0.8735 | 0.0125 |
| PROTSEED | 0.4962 | 0.4071 | 0.3764 | 1.5387 | −6.3829 | 0.6244 | 0.1463 | 10.5301 | 0.4113 | 0.8252 | 0.0557 |
| RFDiffusion | 0.6932 | 0.5728 | 0.0812 | 2.3412 | −6.7446 | 0.6657 | 0.1083 | 20.6224 | 0.6507 | 0.5834 | 0.0239 |
| Chroma | 0.6546 | 0.5163 | 0.4579 | 2.5325 | −6.7258 | 0.7116 | **0.2385** | 9.5328 | 0.6296 | 0.6422 | 0.0968 |
| FADiff | 0.6508 | 0.5351 | 0.3342 | 1.9848 | −6.5924 | 0.6571 | 0.1126 | 7.8516 | 0.4376 | 0.7567 | 0.0731 |
| RFDiffusionAA | 0.7042 | 0.5416 | 0.1134 | 2.5808 | −6.5233 | 0.6732 | 0.1257 | 19.5187 | 0.6461 | 0.6149 | 0.0361 |
| Proteus | 0.6944 | 0.5697 | 0.3926 | 2.1407 | −6.4613 | 0.6816 | 0.1718 | 10.6912 | **0.6716** | 0.6131 | 0.0753 |
| Proteina | 0.7213 | 0.6328 | 0.4583 | 2.4592 | −6.3522 | 0.6709 | 0.1632 | 7.2409 | 0.6542 | **0.5507** | 0.0955 |
| **Ours** | **0.8848** | **0.7160** | **0.5041** | **2.9168** | **−6.9303** | **0.7334** | 0.1861 | **6.9923** | 0.4731 | 0.6739 | **0.1195** |
| Improv. | +23% | +13% | +10% | +13% | +3% | +3% | - | +3% | - | - | +23% |

Table 2: $k$cat comparison across EC families on EnzyBind.

| EC Family | 1.1 | 1.6 | 1.14 | 2.1 | 2.3 | 2.5 | 2.7 | 3.1 | 3.2 | 3.4 | 3.5 | 3.6 | 4.1 | 4.2 | 5.6 | 5.99 | 6.2 | Avg. |
|---|---|---|---|---|---|---|---|---|---|---|---|---|---|---|---|---|---|---|
| EnzyGen | 3.52 | 1.63 | 1.09 | 1.18 | 1.36 | 1.18 | 1.29 | 1.29 | 1.15 | 1.34 | 2.34 | 1.91 | 1.47 | 1.15 | 2.97 | 1.52 | 2.42 | 1.41 |
| PROTSEED | 1.14 | 1.58 | 2.26 | 1.16 | 2.32 | 2.31 | 1.17 | 1.43 | 2.02 | 1.82 | 1.11 | 1.61 | 2.23 | 1.47 | 1.74 | 1.68 | 1.70 | 1.54 |
| RFDiffusion | **5.12** | 3.96 | 2.03 | 2.13 | **2.32** | 1.83 | 1.88 | 2.37 | 2.86 | 2.09 | 3.62 | **3.00** | 4.37 | 2.26 | 2.36 | 1.85 | 2.52 | 2.34 |
| Chroma | 3.79 | 3.28 | 2.59 | 1.79 | 1.82 | **2.62** | **2.76** | **3.70** | 1.84 | 1.70 | 3.26 | 2.20 | 2.92 | 2.50 | **3.18** | 1.48 | 2.20 | 2.53 |
| FADiff | 2.75 | 2.90 | **3.20** | **2.70** | 2.05 | 1.19 | 2.21 | 1.75 | 1.76 | 1.91 | 1.21 | 1.98 | 1.52 | 1.52 | 2.45 | 1.39 | 1.97 | 1.98 |
| Ours | 4.32 | **6.48** | 2.09 | 2.27 | 1.83 | 1.52 | 2.07 | 2.63 | **3.19** | **3.84** | **3.75** | 1.97 | **5.37** | 2.71 | 3.22 | 1.88 | 2.67 | $2.92^{+15.4\%}$ |

CLEAN [92], a sequence-based model with over 90% accuracy across benchmarks. We compute the EC match rate as the proportion of generated enzymes that share the same EC number as their native counterparts. To further evaluate catalytic performance, we predict the catalytic rate constant ($k$cat) using UniKP [93], which takes both sequence and substrate as input. We also evaluate substrate specificity through two metrics: binding affinity, calculated via Gnina [94] (lower is better), and the ESP score from EnzyGen [95], where higher values indicate stronger enzyme-substrate interactions. To provide grounded assessments, we report amino acid recovery (AAR) and RMSD relative to native structures. In addition to performance, we also report Diversity and Novelty.

Finally, we define a success rate to capture practical multi-objective design goals. A generated enzyme backbone is considered successful if it meets all of the following: (i) matches the native enzyme's EC number (EC Match Rate), (ii) scTM > 0.5, (iii) scRMSD < 2Å, and (iv) shows better binding affinity than its native counterpart. Further metric details are provided in App. F.2.

## 5.2 Main Results

Table 1 presents the performance of EnzyControl compared to baseline models on both structural and functional metrics. EnzyControl significantly outperforms existing methods in structural quality. Notably, 0.7160 of its generated enzyme backbones are designable, compared to only 0.6328 for the second-best model, Proteina. It also achieves a substantially higher proportion of structures with scTM > 0.5, showing better self-consistency. On functional metrics, EnzyControl achieves an EC match rate of 0.5041 and a catalytic constant ($k$cat) of 2.9168, outperforming the second-best model by 10% and 15%, respectively. This suggests that EnzyControl not only aligns well with the intended enzymatic functions but also generates enzymes with superior catalytic efficiency. Moreover, it consistently produces enzymes with stronger substrate binding affinity and higher ESP score, achieving 2.8% and 3.1% improvements in these metrics compared to the second-best method.

Despite its strong performances in structural and functional evaluation, EnzyControl lags behind RFDiffusion and Chroma in diversity and novelty. This is due to the larger and more heterogeneous training sets used by those models, which may inflate diversity and novelty scores relative to our benchmark, as also noted in FoldFlow. However, given our primary goal of designing highly specific and functional enzymes, we prioritize performance on structural and functional metrics over diversity and novelty. We further provide evaluation results across different EC families (*cf.* Table 2 and 3). The enzymes generated by EnzyControl show higher catalytic efficiency and stronger binding to their

Table 3: Binding affinity comparison across EC families on EnzyBind.

| EC Family | 1.1 | 1.6 | 1.14 | 2.1 | 2.3 | 2.5 | 2.7 | 3.1 | 3.2 | 3.4 | 3.5 | 3.6 | 4.1 | 4.2 | 5.6 | 5.99 | 6.2 | Avg. |
|---|---|---|---|---|---|---|---|---|---|---|---|---|---|---|---|---|---|---|
| EnzyGen | -6.43 | -6.79 | -6.85 | -6.45 | **-5.09** | -7.44 | -7.62 | -6.15 | -6.48 | -6.94 | -5.71 | -6.33 | -5.93 | -5.58 | -6.52 | -6.72 | -5.00 | -6.75 |
| PROTSEED | -5.28 | -6.25 | -6.69 | -6.43 | -6.14 | -6.36 | -6.35 | -6.28 | -6.81 | -6.24 | -6.53 | -7.63 | -7.62 | -4.55 | -5.99 | -7.45 | -7.45 | -6.38 |
| RFDiffusion | -7.37 | -5.69 | -7.14 | -6.48 | -7.65 | **-5.52** | -6.56 | -6.07 | -6.47 | -7.45 | -7.11 | -7.21 | -6.03 | **-7.41** | -6.24 | -6.82 | -7.48 | -6.74 |
| Chroma | **-4.68** | -7.36 | **-6.13** | **-8.94** | -6.62 | -5.44 | -7.63 | -5.89 | -6.72 | **-6.35** | -5.89 | -7.63 | -5.69 | -6.60 | -5.92 | **-4.97** | -6.57 | -6.73 |
| FADiff | -7.44 | -7.28 | -6.42 | -6.69 | -6.86 | -6.74 | -7.26 | -6.69 | -6.15 | -6.38 | **-5.13** | -5.63 | -7.15 | -7.25 | -5.83 | -5.14 | -5.08 | -6.59 |
| Ours | **-8.78** | **-7.74** | **-9.26** | -6.36 | **-8.11** | -7.12 | **-7.21** | **-6.95** | **-6.86** | -6.55 | -6.05 | **-5.72** | **-5.90** | -5.56 | **-7.39** | **-7.84** | **-10.18** | **-6.93**+2.7% |

Table 4: Effect of motif residue perturbation on design performance. Perturbation rate indicates the fraction of motif residues replaced with random amino acids.

| Perturbation Rate | > 0.5 scTM | Designability | EC Match Rate | $k$cat | Binding Affinity | ESP Score |
|---|---|---|---|---|---|---|
| 100% | 0.8719 | 0.6863 | 0.4764 | 2.4615 | -6.4361 | 0.7183 |
| 50% | 0.8761 | 0.7023 | 0.4918 | 2.6540 | -6.6105 | 0.7238 |
| 0% | 0.8848 | 0.7160 | 0.5041 | 2.9168 | -6.9303 | 0.7334 |

substrates compared to other methods. These results further demonstrate that EnzyControl produces enzymes that are well-aligned with the functional properties of their EC families.

In addition, since EnzyGen is closely related to our work, we conducted a comparison using the EnzyBench dataset from their study. As shown in Table 6 and 7, our model outperforms EnzyGen in terms of Binding Affinity (with a 3% improvement) and pLDDT, achieving the best performance on these metrics. While our ESP score is slightly lower than that of EnzyGen, the developer of the ESP score metric has noted that a score above 0.6 already indicates strong enzyme-substrate interaction.

Table 7: Comparison with EnzyGen on EnzyBench.

| Model | ESP score | Binding Affinity | pLDDT |
|---|---|---|---|
| PROTSEED | 0.59 | -7.94 | 77.07 |
| RFDiffusion | 0.53 | -8.57 | 83.43 |
| ESM2+EGNN | 0.61 | -8.52 | 85.13 |
| EnzyGen | **0.65** | -9.44 | 87.45 |
| Ours | 0.63 | **-9.76** | **88.37** |

## 5.3 Ablation Study and Analysis

Table 5 presents how performance changes when individual components of our model are removed. We first assess the effect of excluding the MSA input by randomly substituting functional sites with unrelated residues. This results in a clear decline in sample quality, especially in functional metrics, highlighting MSA's role in preserving the biological relevance of the generated backbones. Removing the EnzyAdapter produces a similar degradation in performance. Most metrics show reduced scores, with the largest drop in EC match rate. This suggests the EnzyAdapter is critical for generating enzyme backbones that align with their intended functional profiles.

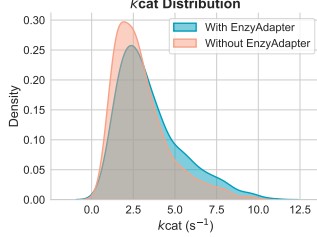

Figure 5: Comparison of $k$cat distribution.

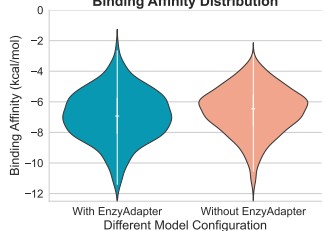

Figure 6: Comparison of binding affinity distribution.

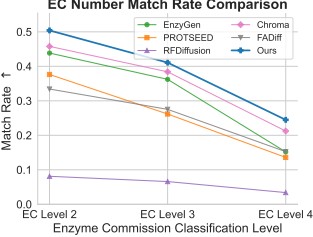

Figure 7: Comparison of EC match rate with different levels.

While Table 5 reports average performance across all samples, our primary objective is to generate enzyme backbones with specific functional properties. To better understand the model's behavior at a finer level, we analyze two key functional indicators: binding affinity and $k$cat, computing their kernel densities (Fig. 5). The x-axis represents $k$cat, and the y-axis shows kernel density estimates forming a continuous curve. A rightward shift shows improved enzymatic efficiency. When EnzyAdapter is removed, the distribution shifts leftward, showing reduced catalytic performance. Fig. 6 illustrates the predicted binding affinity distribution. Here, a lower position on the y-axis means stronger binding

Table 5: Ablating EnzyControl's components. Green denotes relative performance change.

| EnzyAdapter | MSA | >0.5scTM | Designability | EC Match Rate | $k$cat | Binding Affinity | ESP Score |
|---|---|---|---|---|---|---|---|
| ✓ | ✓ | **0.8848** | **0.7160** | **0.5041** | **2.9168** | **-6.9303** | **0.7334** |
| ✗ | ✓ | 0.8748 
 1.32% | 0.7067 
 1.30% | 0.4761 
 5.55% | 2.5833 
 11.43% | -6.5523 
 5.54% | 0.7205 
 1.76% |
| ✓ | ✗ | 0.8719 
 1.46% | 0.6863 
 4.15% | 0.4764 
 5.49% | 2.4615 
 15.61% | -6.4361 
 7.13% | 0.7183 
 2.06% |
| ✗ | ✗ | 0.8684 
 1.85% | 0.6784 
 5.25% | 0.4627 
 8.21% | 2.4492 
 16.03% | -6.3972 
 7.69% | 0.7168 
 2.26% |

Table 6: Binding affinity comparison on EnzyBench. The best-performing results are marked in **bold**.

| EC number | 1.1.1 | 1.14.13 | 1.14.14 | 1.2.1 | 2.1.1 | 2.3.1 | 2.4.1 | 2.4.2 | 2.5.1 | 2.6.1 | 2.7.1 | 2.7.10 | 2.7.11 | 2.7.4 | 2.7.7 |
|---|---|---|---|---|---|---|---|---|---|---|---|---|---|---|---|
| PROTSEED | -6.61 | -4.27 | -10.22 | -6.71 | -8.84 | -9.58 | -7.43 | -10.01 | -7.44 | -5.46 | -8.07 | -9.68 | -10.24 | -11.68 | -8.00 |
| RFDiffusion+IF | -7.11 | -4.70 | -10.74 | -7.21 | -9.61 | -10.04 | -7.93 | -10.64 | -7.84 | -6.19 | -8.55 | -10.60 | -10.44 | -12.18 | -8.50 |
| ESM2+EGNN | -6.66 | -4.81 | -10.73 | -7.02 | -9.57 | -9.98 | -8.61 | -10.90 | -7.95 | -6.43 | -8.79 | -10.23 | -10.75 | -11.31 | -8.47 |
| EnzyGen | -8.44 | -5.10 | -10.34 | -6.95 | -10.05 | -9.89 | **-9.65** | -11.91 | **-9.98** | **-8.05** | **-10.50** | -11.65 | **-12.51** | -11.24 | -8.86 |
| Ours | **-8.58** | **-5.86** | **-11.47** | **-8.29** | **-10.31** | **-10.16** | -7.58 | **-12.20** | -8.32 | -7.72 | -9.47 | **-12.21** | -10.60 | **-12.85** | **-9.12** |

| EC number | 3.1.1 | 3.1.3 | 3.1.4 | 3.2.2 | 3.4.19 | 3.4.21 | 3.5.1 | 3.5.2 | 3.6.1 | 3.6.4 | 3.6.5 | 4.1.1 | 4.2.1 | 4.6.1 | Avg |
|---|---|---|---|---|---|---|---|---|---|---|---|---|---|---|---|
| PROTSEED | -6.01 | -7.20 | -9.16 | -9.53 | -8.79 | -8.67 | -5.19 | -5.44 | -8.57 | -10.11 | -9.37 | -9.74 | -4.38 | -9.11 | -8.12 |
| RFDiffusion+IF | -6.51 | -7.70 | -11.65 | -10.08 | -9.29 | -9.03 | -5.54 | -5.94 | -9.07 | -11.12 | -9.87 | -11.24 | -4.88 | -9.68 | -8.75 |
| ESM2+EGNN | -5.81 | -7.20 | -11.45 | -10.14 | -8.91 | -9.25 | -5.59 | -5.42 | -8.23 | -10.69 | -11.15 | -11.34 | -5.01 | -9.76 | -8.69 |
| EnzyGen | -7.01 | -8.89 | -11.90 | -10.50 | **-10.60** | -10.49 | -6.37 | -6.84 | -9.23 | **-13.10** | -11.35 | -11.03 | -5.51 | -10.64 | -9.61 |
| Ours | **-7.53** | **-9.41** | **-12.20** | **-11.72** | -9.40 | **-10.89** | **-6.65** | **-7.18** | **-9.86** | -12.40 | **-11.60** | **-12.35** | **-6.17** | **-11.13** | **-9.76** [+1.5%] |

(*i.e.,* better performance). The removal of EnzyAdapter notably alters the curve, particularly around -6, underscoring its importance in modeling binding interactions.

To investigate the sensitivity of our method to motif annotation quality, we conduct additional experiments under controlled motif perturbations. As shown in Table 4, even moderate misannotation leads to consistent performance degradation across all functional metrics. Specifically, compared to the unperturbed baseline (0% perturbation), the 50% perturbation setting yields a 2.4% relative decrease in EC match rate, a 9.0% reduction in predicted $k_{\text{cat}}$, and a measurable drop in binding affinity (from $-6.93$ to $-6.61$ kcal/mol). These results confirm two key insights: (1) model performance is highly sensitive to motif fidelity, and (2) our MSA-based annotation strategy is essential for high-quality functional enzyme design. This validates both the design rationale and the importance of accurate functional site annotation in generative modeling.

These results confirm that functional site modeling is critical for designing enzymes with desired activities. However, our current annotations are based on second-level EC categories. To assess alignment at finer granularity, we evaluate the EC match rate at the third and fourth levels, as shown in Fig. 7. Our method, EnzyControl, achieves state-of-the-art performances across all levels, demonstrating superior functional consistency compared to existing baselines.

### 5.4 Quantitative Analysis of Enzyme Function

**Zero-shot.** To evaluate the generalization capability of EnzyControl, we test it on entirely unseen substrates and second-level enzyme categories excluded from the training set. As shown in Fig. 8, the generated enzymes maintained strong binding affinities, averaging –7.0125 kcal/mol across new EC categories and –6.7292 kcal/mol across novel substrates. These results suggest that EnzyControl can effectively design enzymes capable of binding to unfamiliar targets.

**Residue efficiency.** In practical wet-lab settings, engineered enzymes should ideally retain functional activity while minimizing sequence length, as shorter sequences typically enhance gene expression and reduce synthesis cost. To assess this, we analyze the lengths of generated enzyme sequences across different $k$cat intervals. The result is shown in Fig. 9. Compared to the suboptimal baseline, RFDiffusion, EnzyControl consistently produces sequences that are apporximately 30% shorter, while maintaining comparable $k$cat values across all catalytic ranges.

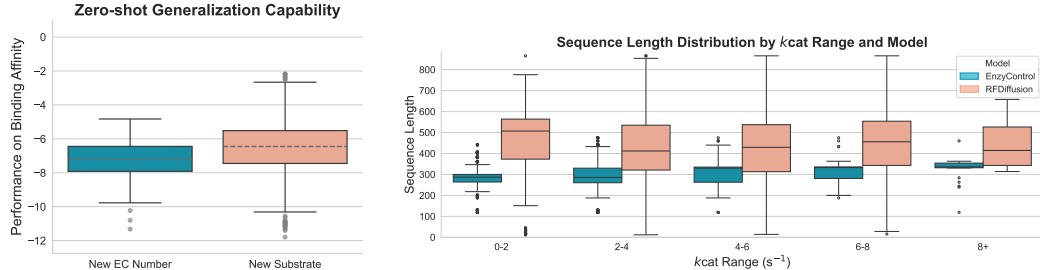

Figure 8: Zero-shot generalization.   Figure 9: Sampled sequence length across different $k$cat range.

**Case study.**   We further validated EnzyControl's performance through a targeted case study using PDB ID: 2cv3 (Fig. 10). The enzyme designed by EnzyControl achieved a binding affinity of –9.78 kcal/mol and a $k$cat of 9.72 s$^{-1}$, representing a 51% improvement in binding affinity and nearly 8× higher catalytic efficiency compared to RFDiffusion. Docking simulations also revealed that the EnzyControl-generated enzyme formed more interaction bonds with the substrate.

Binding Affinity: -6.46 kcal/mol   $k$cat: 1.22 s$^{-1}$   Binding Affinity: -9.78 kcal/mol   $k$cat: 9.72 s$^{-1}$

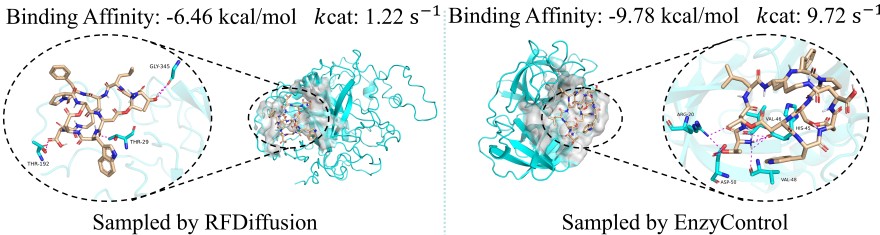

Sampled by RFDiffusion                    Sampled by EnzyControl

Figure 10: Comparison of docking results between EnzyControl and RFDiffusion on the 2cv3 enzyme. EnzyControl achieves better substrate-specificity than RFDiffusion. Additionally, the EnzyControl-generated sample form more interaction bonds compared to the enzyme generated by RFDiffusion.

# 6   Conclusion

In this work, we introduce EnzyControl, a method for generating enzyme backbones tailored to small molecule substrates. Unlike prior approaches, EnzyControl integrates conserved functional motifs—identified through MSA—into the design process, helping preserve key catalytic features. It also uses substrate information to guide backbone refinement, improving binding specificity. To evaluate design quality, we constructed a new benchmark EnzyBind and compare EnzyControl against existing methods. Our results show that EnzyControl achieves leading performance across both structural and functional metrics. Additional analyses demonstrate EnzyControl's strong generalization capabilities, including zero-shot performance, further highlighting its effectiveness for functional enzyme design.

## Acknowledgements

This work was supported in part by the Ministry of Education (MOE T1251RES2309 and MOE T2EP20125-0039), the Agency for Science, Technology and Research (A*STAR H25J6a0034), the National Natural Science Foundation of China (No. 62372375), the Shaanxi Province Key R&D Program (No. 2023-YBSF-114) and the Practice and Innovation Funds for Graduate Students of Northwestern Polytechnical University (PF2024080).

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

# A    Limitations

This work focuses exclusively on generating enzyme backbones, without modeling the specific conformations these backbones adopt when binding to substrates, which is highlighted by AtomicFlow [84]. Additionally, while our method, EnzyControl, produces diverse backbone samples, it struggles to balance diversity with designability. Enhancing designability without sacrificing diversity remains an open challenge and a direction for future work.

# B    More Experimental Results

Due to space limitations, Sec. 5.2 only reports overall test results without distinguishing between enzyme families. To provide a more detailed analysis, Table 9 and 10 present evaluation results broken down by EC family. Additionally, we include functional descriptions for each EC family that appears in the test set, sourced from Expasy[2], as shown in Table 8.

We further provide additional discussions and exploratory results to highlight the extensibility of our framework to more complex biochemical systems, including multimeric assemblies, multi-substrate reactions, and docking-aware design strategies.

**Discussion 1: Extension to Multimeric and Allosteric Systems.** Our current framework focuses on single-chain enzyme scaffolds, which simplifies sequence–structure mapping but limits applicability to multimeric or complex allosteric systems. To explore potential extensibility, we experimented with a post-hoc multimeric assembly pipeline. Specifically, we first generate a single-chain enzyme using our method, and subsequently apply the RFDiffusion binder design module to create a complementary partner that binds to the designed enzyme surface. This strategy enables the construction of multi-chain complexes without modifying the core architecture.

**Original enzyme backbone:**

```
MELPKRRIRLLVLYTPEVEAGPLADPAKREAHIREVVAKVNELLKPFNIEIVLVDIISIGSNYDVDFSAPCEALRAQLEALVAT
KLKKEIDFDMAVVFGGESLAPCIEGFAALGADISTGRGVALAVLDPSDAEADARAVAAQILRLLGVTAPPERRVGPNGGDEDGV
VWGEDGVEESLAWSLEQLRRYFEEHQPAEYLLPP
```

**Designed binder:**

```
SGLERWKEIDENNQWEELTKELLAKQVYRPETNAATGATIIATGPAGAELGAALRAAYGPDPATLVGGVLPRPTTTGIGYAFLG
GVQTPEELARIARLLVSDPTAAVAAYVMTAEDGRIHWDEAAGRAWLAE
```

Preliminary results indicate stable complex formation between the designed enzyme and its binder, suggesting that our framework can be naturally extended to multimeric contexts through modular assembly.

**Discussion 2: Toward Multi-Substrate Enzyme Generation.** While our current model conditions enzyme generation on a single substrate molecule, many natural enzymes interact with multiple substrates or cofactors. To address this, we propose a flexible extension based on substrate-guided representation aggregation. For each substrate, the EnzyAdapter generates a distinct substrate-aware embedding; these embeddings are then aggregated and passed to the Transformer backbone generator. This allows the model to capture multiple substrate interaction contexts simultaneously.

**Example sequence:**

```
LSPEEIEEIKANNQWAERTAALDKTVTLNPSLTLGDWTVDNTGGLDDPDAATRLCRGTIDLATGKIGSGGSVGEKDGGVTIGGL
SLGVEEDGVLHGYLAEISASGATVRVPVRPDDTYRDLAARAQAQLGTSSDAATGATLTLTDIEVRNVGFIITASSA
```

**Substrate 1 (binding affinity = -6.62):**

```
COC(=O)c1c(OC/C=C/c2ccc(c(c2)c2onc(c2)C(=O)O)F)cccc1O
```

**Substrate 2 (binding affinity = -6.4):**

```
OCc1ccc(c(c1)N(c1ccnc(n1)Nc1cc(cc(c1)S(=O)(=O)C)N1CCOCC1)C)C
```

This multi-substrate extension represents a promising direction for broadening the model's biochemical scope, enabling support for multi-step or cofactor-dependent catalytic reactions.

---

[2]https://enzyme.expasy.org/

**Discussion 3: Docking-Aware Optimization Strategies.** In the current version, the substrate is included as an embedding rather than through explicit spatial modeling. To improve docking compatibility, we explored two substrate-aware optimization strategies:

- **Sampling-based selection:** Multiple enzyme backbones are generated per substrate; docking is performed on all candidates, and the structure with the best predicted binding affinity is selected.

- **Motif-branching beam search:** Starting from the annotated catalytic motif, we stochastically extend short N- and C-terminal fragments to create diverse partial scaffolds. Each candidate is completed and docked with the substrate, and the best-scoring motif variant is used as the seed for further generation.

A prototype implementation of the sampling-based approach yielded improved docking affinity:

**Before optimization (binding affinity = -6.92):**

```
MKVFSPALDNPEYYAGILSPEQVKELVALGFTVYILGREHPKSKFTMAELEAAGAVIVKSLEELKGKHDLVLLSVPPGLDDKTR
LPIDTIKKGAIVIGRMKAKTNPEILKALAERGLTVFDMELISPENCDPAMNVVDALGEHVGKVAVRLAKELSSKPFARKETADG
VIPAKKVLVLGWGTAGAAAAREAIALGAEVYVWDIDPEARAVAEAIGATFIAADAEALAEELEKADVIITTDAKRDGKGVVVLS
EEDVKKLKPDSVIVDTTVEDGGACPLAKAGEVVEFNGVKIVGKKNLDSLAPAESTAAYSQCMLNFIKPLVGKGDGELKIDMSRP
CVKDTLVVYNGKIKSKLE
```

**After optimization (binding affinity = -8.38):**

```
MKIFSYALKNPDVYAGILSPEQVKELVALGFEVYISGFEHPKSSFTMEELKAAGATIVDTLEELKGKHDIVLTSVPPGLDNTTA
LPVDTIKPGAILIGRLNAERNPEIIKALAARNLTAFDLERISKDKCPAETNVVDALGKEIGKVAVELAKELSSKPFAAEETADG
LIPAKKVLVLGMGTTGASAAREAIKLGAEVYMYDINPEAKKIAEEIGATFIEEGEEALAAVLKEADVIICTDAMKDGKGLVVLS
AEDVKTLKPDSVIVDTTVERGGACPLAKPGEVVEFEGVKIVGKKNLDSLNPAASQKAFSKCMLNFIKPLVNKGDGELKLNMSDP
CVKDTLVCYKGKIVSPME
```

These strategies demonstrate that docking-guided optimization can substantially improve substrate compatibility, and integrating such mechanisms into the main generative loop represents an important future direction.

Table 8: EC number and corresponding enzyme functions appeared in the testset.

| EC Number | Function |
|---|---|
| 1.1 | Acting on the CH-OH group of donors |
| 1.6 | Acting on NADH or NADPH |
| 1.14 | Acting on paired donors, with incorporation or reduction of molecular oxygen. |
| 2.1 | Transferring one-carbon groups |
| 2.3 | Acyltransferases |
| 2.5 | Transferring alkyl or aryl groups, other than methyl groups |
| 2.7 | Transferring phosphorus-containing groups |
| 3.1 | Acting on ester bonds |
| 3.2 | Glycosylases |
| 3.4 | Acting on peptide bonds (peptidases) |
| 3.5 | Acting on carbon-nitrogen bonds, other than peptide bonds |
| 3.6 | Acting on acid anhydrides |
| 4.1 | Carbon–carbon lyases |
| 4.2 | Carbon–oxygen lyases |
| 5.6 | Isomerases altering macromolecular conformation |
| 5.99 | Other isomerases |
| 6.2 | Forming carbon–sulfur bonds |

## C   Enzyme Commission number

The Enzyme Commission number (EC number) is a numerical classification scheme for enzymes, based on the chemical reactions they catalyze. As a system of enzyme nomenclature, every EC number is associated with a recommended name for the corresponding enzyme-catalyzed reaction. EC numbers do not specify enzymes but enzyme-catalyzed reactions. If different enzymes (for instance from different organisms) catalyze the same reaction, then they receive the same EC number.

Table 9: pLDDT comparison on EnzyBench. The best-performing results are marked in **bold**.

| EC number | 1.1.1 | 1.14.13 | 1.14.14 | 1.2.1 | 2.1.1 | 2.3.1 | 2.4.1 | 2.4.2 | 2.5.1 | 2.6.1 | 2.7.1 | 2.7.10 | 2.7.11 | 2.7.4 | 2.7.7 |
|---|---|---|---|---|---|---|---|---|---|---|---|---|---|---|---|
| PROTSEED | 77.10 | 71.19 | 74.24 | 78.67 | 77.40 | 74.54 | 75.18 | 77.11 | 74.79 | 75.55 | 75.90 | 81.05 | 74.05 | 76.50 | 78.00 |
| RFDiffusion+IF | 82.47 | 81.12 | 89.32 | 82.04 | 82.49 | 85.14 | 85.61 | 81.13 | 86.25 | 81.60 | 87.51 | 86.75 | 85.91 | 88.30 | 81.25 |
| ESM2+EGNN | 90.67 | 90.93 | 90.30 | 87.67 | 79.40 | 84.78 | 84.80 | 84.56 | 90.21 | 87.47 | 83.52 | 88.92 | 85.59 | 90.09 | 81.80 |
| EnzyGen | **91.86** | **93.02** | 92.70 | **91.99** | 83.47 | 87.71 | 92.81 | 87.02 | 89.69 | 89.20 | 85.19 | 87.55 | 87.64 | 91.81 | 83.75 |
| Ours | 91.64 | 90.33 | **93.85** | 89.63 | **85.46** | **88.75** | **93.92** | **88.94** | **91.22** | **90.51** | **88.76** | **89.03** | **88.39** | **92.45** | **86.62** |

| EC number | 3.1.1 | 3.1.3 | 3.1.4 | 3.2.2 | 3.4.19 | 3.4.21 | 3.5.1 | 3.5.2 | 3.6.1 | 3.6.4 | 3.6.5 | 4.1.1 | 4.2.1 | 4.6.1 | Avg |
|---|---|---|---|---|---|---|---|---|---|---|---|---|---|---|---|
| PROTSEED | 76.29 | 77.89 | 75.75 | 78.76 | 73.56 | 82.40 | 76.70 | 75.90 | 75.16 | 74.62 | 83.46 | 76.36 | 78.87 | 83.31 | 76.91 |
| RFDiffusion+IF | 80.01 | 81.59 | 81.22 | **92.04** | **89.72** | 77.20 | 84.05 | 85.47 | 71.35 | 82.87 | 84.49 | 81.31 | 79.02 | 76.11 | 83.22 |
| ESM2+EGNN | 87.27 | **87.05** | 85.50 | 72.23 | 71.31 | 82.62 | 83.48 | 84.96 | 73.34 | 80.77 | 87.72 | 89.70 | 85.48 | | 84.86 |
| EnzyGen | 89.79 | 85.40 | **89.68** | 74.44 | 77.14 | 89.11 | 86.70 | 89.80 | 85.98 | 76.31 | 84.32 | 85.71 | **91.88** | 87.55 | 87.21 |
| Ours | **90.75** | 82.33 | 83.02 | 87.56 | 72.19 | **90.62** | **87.47** | **90.06** | **87.29** | **84.16** | **86.33** | **88.59** | 90.48 | **89.91** | **88.28** |

Table 10: ESP score comparison on EnzyBench. The best-performing results are marked in **bold**.

| EC number | 1.1.1 | 1.14.13 | 1.14.14 | 1.2.1 | 2.1.1 | 2.3.1 | 2.4.1 | 2.4.2 | 2.5.1 | 2.6.1 | 2.7.1 | 2.7.10 | 2.7.11 | 2.7.4 | 2.7.7 |
|---|---|---|---|---|---|---|---|---|---|---|---|---|---|---|---|
| PROTSEED | 0.54 | 0.24 | 0.39 | 0.57 | 0.83 | 0.52 | 0.29 | 0.75 | 0.58 | 0.45 | **0.77** | 0.88 | 0.81 | 0.78 | 0.69 |
| RFDiffusion+IF | 0.45 | 0.54 | 0.39 | 0.47 | 0.43 | 0.48 | 0.39 | 0.52 | 0.46 | 0.53 | 0.50 | 0.51 | 0.60 | 0.55 | 0.53 |
| ESM2+EGNN | 0.58 | 0.35 | 0.35 | 0.63 | 0.79 | 0.53 | 0.32 | 0.80 | 0.59 | 0.51 | 0.76 | 0.88 | 0.88 | 0.77 | 0.70 |
| EnzyGen | 0.64 | 0.38 | 0.42 | **0.72** | 0.80 | **0.61** | 0.38 | **0.86** | 0.66 | 0.53 | 0.76 | **0.92** | **0.93** | 0.80 | **0.79** |
| Ours | **0.67** | **0.56** | **0.51** | 0.65 | **0.84** | 0.59 | **0.47** | 0.79 | **0.72** | **0.65** | 0.68 | 0.53 | 0.55 | **0.82** | 0.61 |

| EC number | 3.1.1 | 3.1.3 | 3.1.4 | 3.2.2 | 3.4.19 | 3.4.21 | 3.5.1 | 3.5.2 | 3.6.1 | 3.6.4 | 3.6.5 | 4.1.1 | 4.2.1 | 4.6.1 | Avg |
|---|---|---|---|---|---|---|---|---|---|---|---|---|---|---|---|
| PROTSEED | 0.70 | **0.90** | 0.84 | 0.48 | 0.29 | 0.69 | 0.31 | 0.10 | 0.50 | 0.57 | 0.37 | 0.84 | 0.83 | 0.42 | 0.58 |
| RFDiffusion+IF | 0.33 | 0.61 | 0.62 | 0.49 | 0.62 | 0.45 | 0.47 | 0.44 | 0.55 | 0.63 | 0.59 | 0.59 | 0.84 | 0.45 | 0.52 |
| ESM2+EGNN | 0.71 | 0.78 | 0.82 | 0.43 | 0.22 | 0.56 | 0.35 | 0.11 | 0.61 | 0.73 | 0.37 | 0.81 | 0.89 | 0.54 | 0.60 |
| EnzyGen | **0.76** | 0.62 | **0.88** | 0.47 | 0.26 | 0.73 | 0.40 | 0.14 | 0.66 | **0.78** | 0.40 | 0.80 | **0.93** | 0.57 | **0.64** |
| Ours | 0.41 | 0.44 | 0.65 | **0.51** | **0.63** | **0.77** | **0.59** | **0.53** | **0.69** | 0.75 | **0.66** | **0.84** | 0.56 | **0.60** | 0.63 |

We also provide an illustration of EC number in Fig. 11. These categories are applied by our EnzyControl to guide the enzyme backbone generation with specific functions.

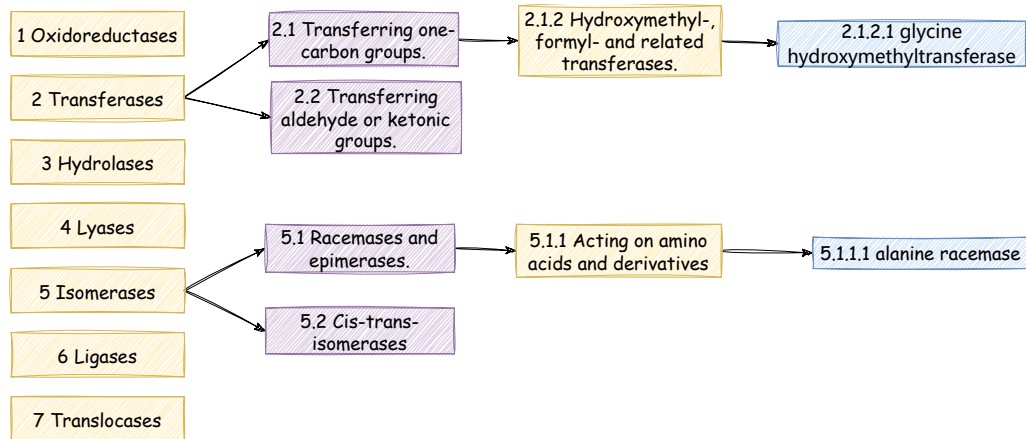

Figure 11: Enzyme Commission (EC) number in BRENDA.

# D More Details on the Dataset

## D.1 Data Licenses

EnzyBind is made available under the Creative Commons Attribution 4.0 International (CC BY 4.0). This license allows users to copy, redistribute, remix, transform, and build upon the dataset for any purpose, including commercial use, provided appropriate credit is given to the creators. A copy of

the license is available at . This dataset is derived from the PDBbind database. PDBbind is a curated database of protein-ligand complexes derived from the Protein Data Bank (PDB). Users must also comply with the licensing terms of the original PDB and PDBbind datasets.

## D.2 Complex Preprocessing

Traditional data-splitting strategies for enzyme datasets often rely on chronological order—training on complexes published before a certain date and testing on those afterward. However, since our objective is to generate enzyme backbones conditioned on desired functions, we adopt a functionally meaningful split based on sequence similarity. Specifically, we use CD-HIT [96] to cluster enzyme sequences and ensure that enzymes in the training and test sets are disjoint. Clusters are then randomly assigned to either training or testing, and enzyme-substrate pairs are sampled accordingly.

Our dataset consists of 11,100 enzyme-substrate complexes, which are first filtered and standardized. We begin by removing complexes that cannot be parsed by the RDKit library [73]. Following the preprocessing pipeline from EquiBind [97], we standardize each molecule using Open Babel [98], correct hydrogen placements on enzymes, and add missing hydrogens with the reduce tool[3].

One remaining challenge is that our model cannot process multi-chain enzymes or symmetric complexes containing repeated enzyme units, as illustrated in Fig. 12. To address this, we retain only the substrate atoms that are within 10Å of any enzyme atom, ensuring that each sample represents a physically relevant interaction while excluding redundant or ambiguous structural data.

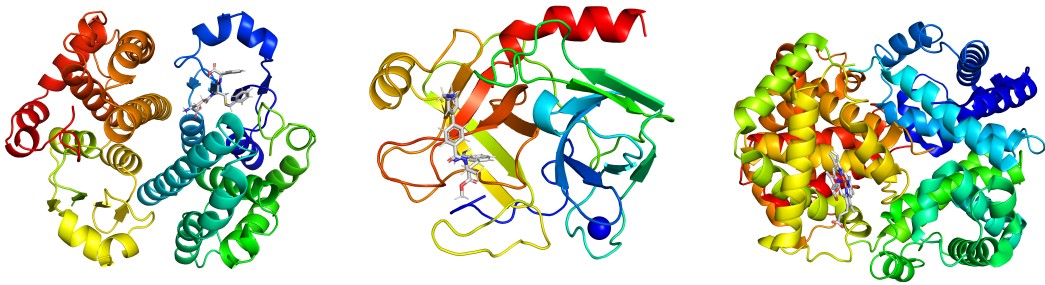

Figure 12: Examples of multi-chain structures and symmetric enzyme complexes.

## D.3 Multiple Sequence Alignment

We identify functional sites in protein sequences using multiple sequence alignment (MSA). As illustrated in Fig. 13, each row represents an enzyme sequence from the same enzyme family, based on the second-level classification in the BRENDA database.

We align these sequences using MAFFT and identify conserved residues by applying an identity threshold $\tau$. Residues that appear consistently across all aligned sequences—such as F, L, and E in our example—are considered functional sites. Following the EnzyGen approach, we set $\tau = 0.3$ in our experiments.

Once functional sites are determined, we encode them as a binary vector with the same length as the original sequence. Each position in the vector is set to 1 if the corresponding amino acid is a functional site, and 0 otherwise.

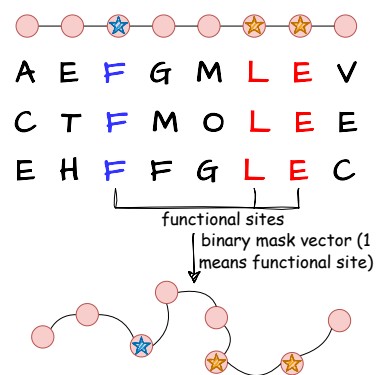

Figure 13: Multiple sequence alignment.

---

[3]https://github.com/rlabduke/reduce

# E More Methodology Details

## E.1 Backbone Frame Representation

We represent each residue's backbone atoms using a local reference frame Fig. 14. As noted in Sec. 4.1, we assume idealized atomic coordinates for the backbone atoms—N, $C_\alpha$, C, O—based on standard chemical bond lengths and angles. To construct a local frame for each residue, we follow the rigid3Point procedure used in AlphaFold2. This method defines a coordinate frame from the backbone atoms using the following steps:

$$
\begin{aligned}
v_1 &= C - C_\alpha, & v_2 &= N - C_\alpha \\
e_1 &= v_1/\|v_2\|, & u_2 &= v_2 - e_1(e_1^T v_2) \\
e_2 &= u_2/\|u_2\| \\
e_3 &= e_1 \times e_2 \\
R &= \text{concat}(e_1, e_2, e_3) \\
x &= C_\alpha \\
T &= (R, x)
\end{aligned}
$$

where the first four lines follow from Gram-Schmidt. The operation of going from coordinates to frames is called atom2frame.

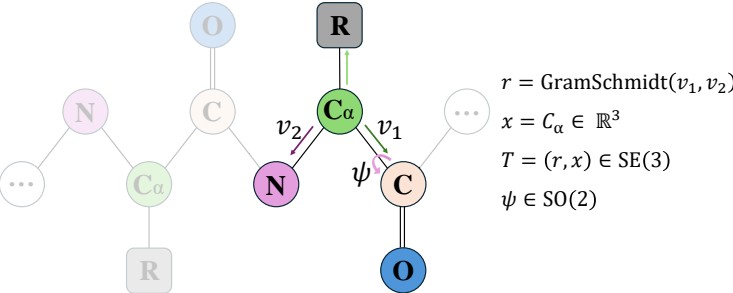

Figure 14: Backbone frame representation.

## E.2 Model Architecture of Projector

The projector consists of two linear layers and a layer normalization (Fig. 15), it can decompose the substrate embedding from the pretrained Uni-Mol encoder and output well-aligned substrate features.

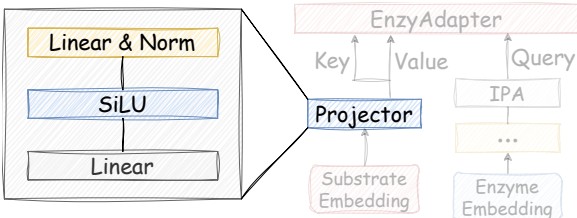

Figure 15: Model architecture of projector.

## E.3 Edge and Backbone Update

**Edge Update.** The edge update step is a critical component of the message-passing mechanism in the Evoformer architecture used by AlphaFold 2. It updates the pairwise edge features by integrating

information from the source and target node embeddings. This process is formally defined as follows:

$$
\begin{aligned}
\boldsymbol{h}_{\text{down}} &= \text{Linear}(\boldsymbol{h}_{k+1}), && \boldsymbol{h}_{\text{down}} \in \mathbb{R}^{D_h/2} \\
\boldsymbol{z}_{\text{in}}^{nm} &= \text{concat}(\boldsymbol{h}_{\text{down}}^{n}, \boldsymbol{h}_{\text{down}}^{m}, \boldsymbol{z}_{k}^{nm}), && \boldsymbol{z}_{\text{in}}^{nm} \in \mathbb{R}^{D_h + D_z} \\
\boldsymbol{z}_{k+1}^{nm} &= \text{LayerNorm}(\text{MLP}(\boldsymbol{z}_{\text{in}}^{nm})), && \boldsymbol{z}_{k+1}^{nm} \in \mathbb{R}^{D_z}
\end{aligned}
\tag{6}
$$

We elaborate on each step below:

- At layer $k + 1$, each node embedding $\boldsymbol{h}_{k+1} \in \mathbb{R}^{D_h}$ is projected to a lower-dimensional representation of size $D_h/2$ using a learned linear transformation. This projection reduces the computational cost of subsequent operations and serves as a bottleneck that encourages the model to extract the most salient features for edge-level reasoning. Notably, this transformation does not include a non-linear activation function.

- To construct the input to the edge update MLP for the residue pair $(n, m)$, the model concatenates three components: the down-projected source node embedding $\boldsymbol{h}_{\text{down}}^{n}$, the down-projected target node embedding $\boldsymbol{h}_{\text{down}}^{m}$, and the current edge feature vector $\boldsymbol{z}_{k}^{nm}$. This combined representation $\boldsymbol{z}_{\text{in}}^{nm}$ lies in $\mathbb{R}^{D_h + D_z}$ and captures both contextual and pairwise information relevant to the interaction between residues $n$ and $m$.

- The combined vector $\boldsymbol{z}_{\text{in}}^{nm}$ is passed through a multi-layer perceptron (MLP), which typically consists of multiple fully connected layers with non-linear activation functions such as ReLU or GELU. The MLP outputs an updated edge embedding $\boldsymbol{z}_{k+1}^{nm} \in \mathbb{R}^{D_z}$. A Layer Normalization operation is applied afterward to stabilize the learning dynamics and ensure consistent feature scaling across the embedding dimension.

**Backbone Update.** The backbone update step in AlphaFold2 is responsible for refining the 3D position and orientation of each residue's local frame using a learnable rigid-body transformation. This transformation is modeled as an element of the special Euclidean group SE(3), combining both rotation and translation. The update proceeds through the following sequence of operations:

$$
\begin{aligned}
b, c, d, x_{\text{update}} &= \text{Linear}(h_k) \\
(a, b, c, d) &= (1, b, c, d) / \sqrt{1 + b^2 + c^2 + d^2} \\
R_{\text{update}} &= \begin{pmatrix} a^2 + b^2 - c^2 - d^2 & 2bc - 2ad & 2bd + 2ac \\ 2bc + 2ad & a^2 - b^2 + c^2 - d^2 & 2cd - 2ab \\ 2bd - 2ac & 2cd + 2ab & a^2 - b^2 - c^2 + d^2 \end{pmatrix} \\
T_{\text{update}} &= (R_{\text{update}}, x_{\text{update}}) \\
T_{k+1} &= T_k \cdot T_{\text{update}}
\end{aligned}
\tag{7}
$$

We elaborate on each step below:

- A linear transformation is applied to the node embedding $h_k \in \mathbb{R}^{D_h}$, producing three scalar values $b, c, d \in \mathbb{R}$ and a translation vector $x_{\text{update}} \in \mathbb{R}^3$. These values parameterize a spatial transformation to be applied to the current residue frame.

- The scalar 1 is prepended to $(b, c, d)$ to construct a 4-dimensional vector $(a, b, c, d)$, which is then normalized to unit norm. This vector forms a unit quaternion, a robust and differentiable representation of a 3D rotation.

- The unit quaternion is converted into a $3 \times 3$ rotation matrix $R_{\text{update}} \in \text{SO}(3)$ using a closed-form expression. This guarantees orthonormality and preserves the group structure of the transformation.

- The rotation $R_{\text{update}}$ is combined with the translation vector $x_{\text{update}}$ to form a rigid-body transformation $T_{\text{update}} \in \text{SE}(3)$, representing a learned update in 3D space.

- Finally, the transformation $T_{\text{update}}$ is applied to the current residue frame $T_k$ by composition, yielding the updated frame $T_{k+1}$. This results in a new position and orientation for the residue, enabling progressive refinement of backbone geometry across Evoformer layers.

During generation, we condition the generation process on known structural motifs which are provided are treated as fixed anchors and stored as $x_1 = \{\text{trans}_1, \text{rot}_1\}$. A binary mask determines which parts of the structure are generated and which parts are clamped to the known motif. At each denoising step, we overwrite the motif region in the predicted structure with its true value from $x_1$, ensuring that motif geometry remains unchanged throughout the sampling process. This design enables consistent integration of known substructures while flexibly generating surrounding regions. The pesudocode is shown in Alg. 1.

---

**Algorithm 1** Inference

**Require:** annotated motifs $x_1 = \{\text{trans}_1, \text{rot}_1\}$, model parameters $\theta$, schedule $t$, motif_mask $\in \{0,1\}^n$, number of steps $m$.
1: Initialize noisy sample $x_0 \leftarrow q(x)$, *e.g.*, Gaussian translation and IGSO(3) rotation
2: $x_t \leftarrow x_0$
3: **for** $i = 0$ to $m - 1$ **do**
4:    $x_t, t_i, x_1, \text{motif\_mask}, \text{substrate} \leftarrow \text{DataLoader}$
5:    Predict vector field: $\Delta x \leftarrow \text{model}(x_t; t_i; \theta)$
6:    Euler step updating: $x_{t+1} \leftarrow x_t + \Delta t \cdot \Delta x$
7:    Overwrite motif structure: $x_{t+1} \leftarrow x_{t+1} \cdot \text{motif\_mask} + x_t \cdot (1 - \text{motif\_mask})$
8: **end for**
9: **return** Trajectory $x_0 \rightarrow x_1$

---

# F More Details on Experimental Settings

## F.1 Additional Training Details

We adopt Low-Rank Adaptation (LoRA) with a rank of $r = 16$ and a scaling factor $\alpha = 32$, targeting key linear projection modules across attention and embedding components, as specified in Table 11. The node and edge embeddings are configured with dimensionalities of 256 and 128, respectively.

Our model supports a maximum of 2000 residues and embeds 1000 discrete timesteps using both sinusoidal and learned positional encodings. Node-level features include spatial coordinates, timestep embeddings, and optional chain-level signals. For edge features, we employ relative position encoding, discretized into 22 bins, and include diffusion-specific masks and self-conditioning mechanisms to enhance robustness.

The IPA module comprises six stacked blocks with multi-head attention (8 heads), point-based QK and V projections, and a lightweight sequence-level Transformer consisting of 2 layers with 4 heads each. These configurations were selected based on empirical validation to balance computational efficiency with modeling capacity.

## F.2 Evaluation Details

This section details the computation of each evaluation metric used in our study.

To evaluate self-consistency, we measure the structural similarity between the generated protein backbones and the all-atom structures predicted by ESMFold. Specifically, we compute the TM-score and RMSD between the two. TM-scores are calculated using the tmtools, while RMSD is computed after structural alignment following the procedure described in FrameFlow.

We assess two aspects of enzyme properties: EC number classification and catalytic efficiency ($k$cat). Both are predicted using pretrained models. For EC number prediction, we use CLEAN, which takes only the amino acid sequence as input. For kcat prediction, we input both the sequence and the substrate's SMILES representation into a separate predictive model. We define the EC match rate as the proportion of samples whose predicted EC number matches that of the corresponding native enzyme. For $k$cat, we report the average predicted value across all samples.

Substrate binding is evaluated using two methods. First, we perform docking simulations with GNINA to assess how well each enzyme binds to a given molecule. We treat the entire enzyme structure as the search space for docking. Second, we compute the ESP score using the pretrained ESP model, which takes both the sequence and substrate as inputs. The final ESP score is the mean value across all samples.

Diversity measures the proportion of unique structural clusters among generated enzymes, while Novelty reflects how dissimilar the generated structures are from native ones, computed as the average

Table 11: Model configuration and hyperparameter settings.

| Module | Parameter | Value | Description |
|---|---|---|---|
| LoRA Settings | lora_r | 16 | Rank of low-rank matrices in LoRA. |
| | lora_alpha | 32 | Scaling factor for LoRA adaptation. |
| | lora_dropout | 0.0 | Dropout applied to LoRA layers. |
| | lora_bias | "none" | Whether LoRA includes bias terms. |
| | lora_target_modules | List of 12 modules | Target modules for LoRA adaptation. |
| Embedding Dimensions | node_embed_size | 256 | Dimensionality of node embeddings. |
| | edge_embed_size | 128 | Dimensionality of edge embeddings. |
| Node Features | c_s | 256 | Size of node feature representation. |
| | c_pos_emb | 128 | Positional embedding dimensionality. |
| | c_timestep_emb | 128 | Timestep embedding dimensionality. |
| | max_num_res | 2000 | Maximum number of residues. |
| | timestep_int | 1000 | Number of discrete time intervals. |
| | embed_chain | False | Whether to embed chain-level info. |
| Edge Features | single_bias_transition_n | 2 | Number of single bias transitions. |
| | c_s | 256 | Node representation dimension. |
| | c_p | 128 | Edge embedding dimensionality. |
| | relpos_k | 64 | Relative positional embedding size. |
| | feat_dim | 64 | Feature vector dimensionality. |
| | num_bins | 22 | Number of distance bins. |
| | self_condition | True | Whether to apply self-conditioning. |
| IPA Module | c_s | 256 | Input node feature dimension. |
| | c_z | 128 | Input edge feature dimension. |
| | c_hidden | 128 | Hidden dimension of IPA module. |
| | no_heads | 8 | Number of attention heads. |
| | no_qk_points | 8 | Number of QK reference points. |
| | no_v_points | 12 | Number of V reference points. |
| | seq_tfmr_num_heads | 4 | Heads in sequence-level Transformer. |
| | seq_tfmr_num_layers | 2 | Layers in sequence-level Transformer. |
| | num_blocks | 6 | Number of IPA module blocks. |

of the maximum TM-scores between each designable enzyme and all native proteins—lower values imply more novel designs. We use Foldseek. Diversity is measured by clustering the generated proteins with Foldseek and calculating the ratio of the number of clusters to the total number of samples. Novelty is defined as the average of the maximum TM-scores between each generated enzyme and all native proteins. Lower scores indicate greater novelty, as they reflect less structural similarity to known proteins.

While these metrics are model-based, they are grounded in experimentally validated frameworks and have demonstrated predictive fidelity in wet-lab settings:

- **EC number prediction** is performed using CLEAN [92], a sequence-based model trained and benchmarked on large-scale enzymatic datasets. CLEAN achieves over 90% accuracy on standard benchmarks and has been experimentally validated on real enzymes such as MJ1651 and SsFIA, with demonstrated capability to predict novel EC assignments.

- **Catalytic rate constant** ($k_{cat}$) is estimated via UniKP [93], which is trained on experimentally measured kinetic data. Its predictions have been corroborated by wet-lab validation on tyrosine ammonia lyase (TAL), confirming its reliability in capturing catalytic efficiency.

- **Enzyme–substrate interaction strength** is quantified using the ESP score from EnzyGen [95], which incorporates statistical testing to ensure interpretability and confidence in its predictions.

- **Binding affinity** is computed using Gnina [94], a physics-based molecular docking tool. This provides an orthogonal, structure-driven validation of substrate compatibility that complements sequence- and learning-based functional metrics.

### F.3 Computational Resource

All experiments were conducted on a high-performance computing node equipped with 4× NVIDIA A100 GPUs (80GB) and dual Intel(R) Xeon(R) Gold 6348 CPUs (2.60GHz, 2 sockets, 28 cores per socket, 112 threads in total).

