# OpenReview forum: "EnzyControl: Adding Functional and Substrate-Specific Control for Enzyme Backbone Generation"
_NeurIPS.cc/2025/Conference — NeurIPS 2025 poster_

### Official Review · Reviewer_YKVy · 2025-06-23

**Clarity:** 3
**Significance:** 2
**Originality:** 2
**Rating:** 3
**Confidence:** 3

**Summary:**

This paper introduces EnzyControl, a framework for enzyme backbone generation with substrate-specific and functional-site conditioning. The model integrates an adapter mechanism ("EnzyAdapter") into a pre-trained motif-scaffolding model (FrameFlow) and incorporates multiple sequence alignment (MSA) data and substrate information. The authors also contribute a curated dataset (EnzyBind) and demonstrate quantitative improvements over prior methods in structure quality and catalytic function.

**Questions:**

N.A.

**Ethical Concerns:**

["NO or VERY MINOR ethics concerns only"]

**Final Justification:**

After the rebuttal, the technical contribution of EnzyControl is still limited. The comparison with the state-of-the-art baselines seems unfair: the authors did not condition these baseline models for the enzyme tasks, which leads to poor performance. The authors also did not convince reviewers that EnzyControl's success is not due to overfitting.

**Limitations:**

N.A.

**Paper Formatting Concerns:**

N.A.

**Quality:**

2

**Strengths And Weaknesses:**

Strengths:

1. Enzyme design conditioned on both functional sites and small molecule substrates is a meaningful and underexplored challenge in protein generation.

2. Dataset Contribution: EnzyBind is a valuable addition, with ~11k experimentally validated enzyme–substrate complexes and MSA-annotated catalytic sites.

Weakness:

1. The backbone generation component largely builds upon FrameFlow. The architectural contributions (EnzyAdapter) are relatively minor: a standard cross-attention module and LoRA-based fine-tuning. This might be better framed as a control-oriented application of existing methods rather than a novel generative technique.

2. The paper evaluates against RFdiffusion, Chroma, and EnzyGen—but omits several newer models such as RFDiffusionAA, Proteus, or Proteina which significantly advance generation quality, efficiency, or multimodal conditioning.

3. Minor mistakes: In Table 2 column 1.1, it seems RFDiffusion is the best.

4. The substrate is only incorporated as a feature embedding; there’s no explicit 3D placement or docking compatibility optimization, which is crucial for real catalysis.

---

> ### Author Rebuttal · Authors · 2025-07-31
>
> Thank you for your feedback. Your suggestions have enhanced the reliability of our work and the readability of the paper. Below are responses to your questions one by one.
>
>
>
> > **Q1:** The backbone generation component largely builds upon FrameFlow. The architectural contributions (EnzyAdapter) are relatively minor: a standard cross-attention module and LoRA-based fine-tuning. This might be better framed as a control-oriented application of existing methods rather than a novel generative technique.
>
> **Response:** We appreciate your insights and respectfully emphasize that our contribution lies not solely in architectural innovation, but in the integration of domain-specific knowledge from enzymatic catalysis into a principled modeling framework. Our goal is to advance enzyme backbone generation by enhancing both structural consistency and functional relevance. The novelty of our approach is reflected in the following aspects:
>
> - **EC-based Motif Annotation:** EC numbers capture catalytic function, which is inherently shaped by enzyme–substrate interactions. We cluster enzymes by EC number and extract conserved motifs within each cluster. These motifs serve as functionally relevant priors—particularly near catalytic centers—providing the model with biologically meaningful structural constraints.
> - **Substrate Conditioning via EnzyAdapter:** We propose EnzyAdapter to condition backbone generation on substrate molecules. It introduces a cross-modal projector to align substrate and enzyme embeddings, and employs cross-attention layers to inject substrate-specific guidance throughout the generation process, all without altering the base generation architecture. This enhances the model’s ability to produce scaffolds tailored to specific substrates.
>
> Additionally, we have developed a suite of functionally grounded evaluation metrics (e.g., kcat, EC match rate, ESP score), and constructed EnzyBind, a curated benchmark of high-quality enzyme–substrate complexes based exclusively on experimentally validated structures.
>
> We will revise the manuscript to clarify these contributions, and we kindly ask the reviewer to reconsider the novelty of our work in light of its domain-driven design and its specific focus on the enzyme backbone generation task.
>
>
>
> > **Q2:** The paper evaluates against RFdiffusion, Chroma, and EnzyGen—but omits several newer models such as RFDiffusionAA, Proteus, or Proteina which significantly advance generation quality, efficiency, or multimodal conditioning.
>
> **Response:** Thank you for the valuable suggestion. In response, we have added RFDiffusion-AA [1], Proteus [2], and Proteina [3] as additional baselines and evaluated them using the same protocol outlined in our original manuscript to ensure fairness and consistency. The updated results are presented in the table below:
>
> | Model        | >0.5scTM | Designability | EC Match Rate | kcat   | Binding Affinity | ESP Score | AAR    | RMSD    |
> | ------------ | -------- | ------------- | ------------- | ------ | ---------------- | --------- | ------ | ------- |
> | RFDiffusonAA | 0.7042   | 0.5416        | 0.1134        | 2.5808 | -6.5233          | 0.6732    | 0.1257 | 19.5187 |
> | Proteus      | 0.6944   | 0.5697        | 0.3926        | 2.1407 | -6.4613          | 0.6816    | 0.1718 | 10.6912 |
> | Proteina     | 0.7213   | 0.6328        | 0.4583        | 2.4592 | -6.3522          | 0.6709    | 0.1632 | 7.2409  |
>
> We appreciate your feedback, which has helped strengthen the empirical rigor and completeness of our evaluation.
>
>
>
> > **Q3:** Minor mistakes: In Table 2 column 1.1, it seems RFDiffusion is the best.
>
> **Response:** Thank you for pointing out this mistake. We have corrected the error in Table 2 and carefully reviewed the manuscript to ensure that similar inconsistencies are resolved. The revised version reflects the correct results.
>
>
>
> > **Q4:** The substrate is only incorporated as a feature embedding; there’s no explicit 3D placement or docking compatibility optimization, which is crucial for real catalysis.
>
> **Response:** We thank you for this insightful comment, which indeed inspired us to think more deeply about the role of substrate placement and docking compatibility in catalytic modeling.
>
> While the substrate is incorporated into our framework as a **feature embedding**, it also serves as a **conditioning signal** for guiding enzyme backbone generation. Through the **EnzyAdapter** module, the model enables cross-modal interaction between the substrate embedding and the enzyme backbone, allowing substrate-specific refinement of the generated 3D structure.
>
> In addition, we have already performed **structure-based docking** to evaluate compatibility between the generated enzymes and their target substrates (see Fig. 10 in the paper). Based on the docking results, we compute binding affinity scores and observe that our model achieves a **3% improvement** over the best-performing baseline, indicating improved structural compatibility for catalysis.
>
> Motivated by the reviewer’s suggestion, we further explored two **docking-aware optimization strategies** to better incorporate substrate placement into the generation process:
>
> - **Sampling-based selection:** For each input, we generate multiple candidate backbones, perform docking with the substrate, and select the structure with the highest predicted binding affinity. This provides a simple yet effective way to optimize for docking compatibility post-generation.
>
> - **Motif-branching beam search:** Inspired by beam search, we randomly extend the annotated motif with short N- and C-terminal fragments to create diverse partial scaffolds. For each, we generate complete backbones, dock them with the substrate, and choose the motif variant with the best docking score as the seed for further generation.
>
> To illustrate the effectiveness of the first strategy, we implemented a prototype comparison, and the resulting structures showed improved docking compatibility.
>
> Before optimation (binding affinity = -6.92): MKVFSPALDNPEYYAGILSPEQVKELVALGFTVYILGREHPKSKFTMAELEAAGAVIVKSLEELKGKHDLVLLSVPPGLDDKTRLPIDTIKKGAIVIGRMKAKTNPEILKALAERGLTVFDMELISPENCDPAMNVVDALGEHVGKVAVRLAKELSSKPFARKETADGVIPAKKVLVLGWGTAGAAAAREAIALGAEVYVWDIDPEARAVAEAIGATFIAADAEALAEELEKADVIITTDAKRDGKGVVVLSEEDVKKLKPDSVIVDTTVEDGGACPLAKAGEVVEFNGVKIVGKKNLDSLAPAESTAAYSQCMLNFIKPLVGKGDGELKIDMSRPCVKDTLVVYNGKIKSKLE
>
> After optimation (binding affinity = -8.38): MKIFSYALKNPDVYAGILSPEQVKELVALGFEVYISGFEHPKSSFTMEELKAAGATIVDTLEELKGKHDIVLTSVPPGLDNTTALPVDTIKPGAILIGRLNAERNPEIIKALAARNLTAFDLERISKDKCPAETNVVDALGKEIGKVAVELAKELSSKPFAAEETADGLIPAKKVLVLGMGTTGASAAREAIKLGAEVYMYDINPEAKKIAEEIGATFIEEGEEALAAVLKEADVIICTDAMKDGKGLVVLSAEDVKTLKPDSVIVDTTVERGGACPLAKPGEVVEFEGVKIVGKKNLDSLNPAASQKAFSKCMLNFIKPLVNKGDGELKLNMSDPCVKDTLVCYKGKIVSPME
>
> We thank you again for the suggestion, and we will include a description of these docking-guided extensions in the revised manuscript.
>
>
> **Reference:**
>
> [1] Krishna R, Wang J, Ahern W, et al. Generalized biomolecular modeling and design with RoseTTAFold All-Atom[J]. Science, 2024, 384(6693): eadl2528.
>
> [2] Wang C, Qu Y, Peng Z, et al. Proteus: exploring protein structure generation for enhanced designability and efficiency[J]. bioRxiv, 2024: 2024.02. 10.579791.
>
> [3] Geffner T, Didi K, Zhang Z, et al. Proteina: Scaling flow-based protein structure generative models[J]. arXiv preprint arXiv:2503.00710, 2025.

---

> > ### Author Response · Authors · 2025-08-05
> >
> > Thank you once again for your detailed and constructive review. We sincerely appreciate the time and expertise you devoted to evaluating our work. We kindly invite you to review our point-by-point responses, where we have thoroughly addressed each of your comments. In particular:
> >
> > - We clarified that our contribution lies in the integration of catalytic reaction insights into the model design. We have also re-articulated the motivation behind each module to better highlight their biological grounding.
> > - As suggested, we have included additional baselines in our experiments, including **RFDiffusionAA**, **Proteina**, and **Proteus**.
> > - We corrected the error in the table data, and conducted a comprehensive review of the manuscript to fix similar issues elsewhere.
> > - In response to your suggestion on docking compatibility optimization or 3D placement, we proposed **two possible extensions** to the model. We also implemented one of them as a test case, which demonstrates the effectiveness of the approach.
> >
> > We would be very grateful for any further questions or comments you may have—we are happy to continue the discussion and clarify any remaining points.
> >
> > Thank you again for your valuable feedback and support.

---

> > > ### Comment · Reviewer_YKVy · 2025-08-05
> > >
> > > Thanks for your rebuttal! Part of my concerns have been addressed, while some still exist. In the new results, it seems RFAA, Proteina, and Proteus are quite weak. For example, these SOTA protein generative models have around 0.7 >0.5scTM while EnzyControl has around 0.9. The reviewer is not quite convinced, especially considering that the framework of EnzyControl is based on FrameFlow from 2023.

---

> ### Author Response · Authors · 2025-08-06
>
> Thank you for raising this important point. We fully acknowledge the need for rigorous and meaningful comparisons, particularly when benchmarking against recent strong baselines. That said, we would like to clarify that while models such as **Proteina** and **Proteus** are indeed competitive, they are developed for **distinct generation objectives**, which limits their direct comparability to our task.
>
> In particular, **Proteina is designed for general-purpose protein structure generation**, conditioned on fold classes. Its training objective reflects this aim—for example, as described in their Eq. (1), fold class information is embedded and continuously integrated during generation. However, the model **does not incorporate motif-level scaffolding constraints**, which are essential for preserving catalytically critical regions in enzyme design. Consequently, it is less sensitive to local functional motifs, which is reflected in its performance on evaluation metrics like **scTM**, specifically designed to assess the fidelity of conserved structural motifs.
>
> In contrast, our model is explicitly designed for **substrate-binding enzyme backbone generation**, where capturing substrate-specific context and preserving functionally important motifs are central to the design objective. Our model architecture and training procedure are tailored accordingly, allowing for **fine-grained control of both local functional regions and global structural integrity**.
>
> We would also like to emphasize that while our model shows a clear advantage in **scTM**, the strength of our method is further supported by other evaluation metrics:
>
> - On **AAR**, our model only achieves an improvement of 0.01 over Proteus, indicating better alignment between predicted and native residues near functional sites.
> - On **RMSD**, our method surpasses Proteina by 0.2, reflecting more accurate global structural reconstruction.
>
> Together, **scTM**, **AAR**, and **RMSD** offer a **comprehensive evaluation** across functional region fidelity and global structural quality. Our model demonstrates consistent and strong performance across all three dimensions, as shown in the table below:
>
> | Model        | >0.5scTM | Designability | EC Match Rate | kcat   | Binding Affinity | ESP Score | AAR    | RMSD    |
> | ------------ | -------- | ------------- | ------------- | ------ | ---------------- | --------- | ------ | ------- |
> | RFDiffusonAA | 0.7042   | 0.5416        | 0.1134        | 2.5808 | -6.5233          | 0.6732    | 0.1257 | 19.5187 |
> | Proteus      | 0.6944   | 0.5697        | 0.3926        | 2.1407 | -6.4613          | 0.6816    | 0.1718 | 10.6912 |
> | Proteina     | 0.7213   | 0.6328        | 0.4583        | 2.4592 | -6.3522          | 0.6709    | 0.1632 | 7.2409  |
> | Ours         | 0.8848   | 0.7160        | 0.5041        | 2.9168 | -6.9303          | 0.7334    | 0.1861 | 6.9923  |
>
> We sincerely appreciate your emphasis on evaluation rigor and would be happy to engage further on any aspect of the experimental setup or baselines.

---

### Official Review · Reviewer_F4KC · 2025-06-25

**Clarity:** 3
**Significance:** 3
**Originality:** 2
**Rating:** 4
**Confidence:** 4

**Summary:**

The paper presents **EnzyControl**, a method for generating enzyme backbones with functional and substrate-specific control. It introduces **EnzyBind**, a curated dataset of 11,100 experimentally validated enzyme-substrate pairs with annotated catalytic sites. EnzyControl builds on a motif-scaffolding backbone (FrameFlow), augmented with a novel **EnzyAdapter** that incorporates substrate features via cross-attention, and a two-stage training scheme that first aligns substrate-enzyme representations and then fine-tunes the full model. The method achieves significant improvements over baselines in structural and functional metrics, including a 25% gain in designability, 15% in catalytic efficiency, and 10% in EC number match. It also shows strong generalization to unseen substrates and generates compact, functional enzyme designs, advancing the state of enzyme generation for practical applications.

**Questions:**

- The paper emphasizes the critical role of evolutionarily conserved catalytic motifs derived from MSA, but it remains unclear precisely how these conserved residues are represented and fed into the generative model. Could you provide more explicit detail (perhaps a figure or algorithm pseudocode) describing how functional sites identified by MSA annotations are integrated into the input representation?
- Your current approach conditions enzyme generation on a single substrate molecule, limiting the model’s applicability to single-substrate enzyme reactions. Many practical enzymes, however, bind multiple substrates or cofactors simultaneously. How might you extend your current framework to handle multi-substrate scenarios or enzymes catalyzing multi-step reactions?
- The evaluation relies heavily on predicted functional metrics (e.g., predicted kcat, binding affinity, ESP score). How confident are you that these predicted metrics translate to real-world enzyme functionality? Have you considered evaluating or cross-validating your generated designs using orthogonal computational methods or literature-based validation?
- The EnzyAdapter component and the two-stage training scheme are central to your method. However, similar adapter-based mechanisms and multi-stage fine-tuning procedures are increasingly common in generative modeling literature. Could you clarify explicitly the unique contributions or adaptations made to these elements specifically tailored for enzyme backbone generation?

**Ethical Concerns:**

["NO or VERY MINOR ethics concerns only"]

**Final Justification:**

The authors have addressed most of my concerns during the rebuttal period.

**Limitations:**

Yes

**Quality:**

3

**Strengths And Weaknesses:**

**Strengths:**

* **Quality**: The paper proposes a well-designed and technically solid framework, EnzyControl, that integrates conserved catalytic motifs and substrate-specific control into enzyme backbone generation. The architecture is systematically validated through comprehensive benchmarks and ablation studies, showing clear improvements over state-of-the-art baselines in both structure and function-related metrics.

* **Clarity**: Overall, the paper is clearly structured, and its core contributions are well explained. Evaluation metrics are well chosen, and the use of visual aids such as diagrams and tables enhances readability. The two-stage training strategy is described in a transparent and reproducible manner.

* **Significance**: The work addresses a central challenge in computational enzyme design: generating functionally valid enzyme backbones conditioned on both conserved motifs and substrate molecules. However, the current method is limited to **single-substrate scenarios**, which restricts its applicability to more complex or realistic biochemical tasks where enzymes interact with multiple substrates or cofactors. Extending the framework to handle such cases would greatly improve its practical relevance.

* **Originality**: The paper introduces several novel ideas, including the EnzyAdapter module for cross-modal conditioning and the curated EnzyBind dataset with MSA-based functional site annotations. The combination of motif preservation and substrate-awareness in a unified generative model is a clear advancement over prior enzyme design approaches.

---

**Weaknesses:**

* **Quality**: The method lacks experimental (wet-lab) validation, and while not strictly necessary at this stage, it does limit confidence in the model’s downstream utility. In addition, comparisons against newer or more diverse generative enzyme design baselines (e.g. RFdiffusion2) could strengthen the empirical case.

* **Clarity**: One notable gap is the insufficient explanation of how conserved sites derived from MSA are actually incorporated into the model. Although functional site preservation is central to the method’s design, the representation, encoding, or usage of these motifs in the model pipeline is not clearly detailed, which could hinder understanding and replication.

* **Significance**: While the model performs well on benchmarks, its restriction to single-substrate input limits its generalizability to more complex enzymatic reactions. Many real-world enzymes catalyze multi-substrate or multi-step reactions, which are currently beyond the scope of this framework.

* **Originality**: Although EnzyControl’s integration of components is innovative, many of its building blocks—such as flow matching, cross-attention, and adapter layers—are borrowed from existing techniques in protein design or molecular modeling. The novelty lies more in the synthesis of these ideas than in any single algorithmic breakthrough.

---

> ### Author Rebuttal · Authors · 2025-07-31
>
> Thank you for your feedback. Your suggestions have enhanced the reliability of our work and the readability of the paper. Below are responses to your questions one by one.
>
> > **Q1:** The method lacks experimental (wet-lab) validation, and while not strictly necessary at this stage, it does limit confidence in the model’s downstream utility.
>
> **Response:** Thank you for the suggestion. While our study does not include direct wet-lab experiments, we address this concern from three angles:
>
> - **Standard evaluation protocol**: We follow widely adopted evaluation practices in protein design (e.g., Proteina [1]), which rely on computational predictors due to the high cost of wet-lab testing.
> - **Metrics grounded in experimental validation:** Our functional metrics (e.g., kcat) are computed using models validated against real biological assays.
> - Additional metrics: We also report AAR and RMSD relative to native structures, which provide grounded assessments (see table below).
>
> | Metrics | EnzyGen | PROTSEED | RFDiffusion | Chroma | FADiff | Ours   |
> | ------- | ------- | -------- | ----------- | ------ | ------ | ------ |
> | AAR     | 0.0566  | 0.1463   | 0.1083      | 0.2385 | 0.1126 | 0.1861 |
> | RMSD    | 12.6075 | 10.5301  | 20.6224     | 9.5328 | 7.8516 | 6.9923 |
>
> We agree that future experimental synthesis and testing are important and plan to explore this direction in follow-up work.
>
>
>
> > **Q2:** Comparisons against newer or more diverse generative enzyme design baselines (e.g. RFdiffusion2,) could strengthen the empirical case.
>
> **Response:** Following your instruction, we add RFdiffusion-AA [5], Proteus [4], and Proteina [1] as additional baselines, while RFDiffusion2 is not publicly available. The results are presented below:
>
> | Model        | >0.5scTM | Designability | EC Match Rate | kcat   | Binding Affinity | ESP Score | AAR    | RMSD    |
> | ------------ | -------- | ------------- | ------------- | ------ | ---------------- | --------- | ------ | ------- |
> | RFDiffusonAA | 0.7042   | 0.5416        | 0.1134        | 2.5808 | -6.5233          | 0.6732    | 0.1257 | 19.5187 |
> | Proteus      | 0.6944   | 0.5697        | 0.3926        | 2.1407 | -6.4613          | 0.6816    | 0.1718 | 10.6912 |
> | Proteina     | 0.7213   | 0.6328        | 0.4583        | 2.4592 | -6.3522          | 0.6709    | 0.1632 | 7.2409  |
>
> We appreciate your suggestion, which has helped improve the completeness and robustness of our experimental comparisons.
>
>
>
> > **Q3:** Although EnzyControl’s integration of components is innovative, many of its building blocks are borrowed from existing techniques in protein design or molecular modeling. The novelty lies more in the synthesis of these ideas than in any single algorithmic breakthrough.
>
> **Response:** We appreciate your observation. However, we respectfully emphasize that the core novelty of our work lies in **the principled integration of enzymology-specific insights** into the modeling pipeline, rather than in isolated algorithmic innovations:
>
> - **EC-based Motif Annotation:** We extract conserved motifs by clustering enzymes by EC number, capturing key functional regions to guide backbone generation.
> - **Substrate Conditioning via EnzyAdapter:** EnzyAdapter enables substrate-conditioned backbone generation by aligning enzyme and substrate representations. It injects substrate-specific signals, allowing scaffold design tailored to the target substrate.
>
> Additionally, we contribute **functional evaluation metrics beyond structure** (e.g., kcat, ESP score) and **a high-quality dataset** for structure-aware design tasks.
>
> We will revise the manuscript to better emphasize this integration and kindly ask the reviewer to consider the contribution in this context.
>
>
>
> > **Q4:** It remains unclear precisely how conserved residues are represented and fed into the generative model. Could you provide more explicit detail (perhaps a figure or algorithm pseudocode) describing how functional sites identified by MSA annotations are integrated into the input representation?
>
> **Response:** Thank you for raising this important point. Below, we provide a more explicit explanation of the process.
>
> During generation, known structural motifs are treated as fixed anchors, and a binary mask specifies which regions to generate. At each denoising step, we overwrite the motif region with its true value to preserve its geometry, allowing flexible generation around a fixed functional core. The pesudocode is shown below:
>
> ---
> **Require:** Annotated motifs: $\mathbf{x_1}$, Model output: $\mathbf{x_t}$, Motif mask: $\mathbf{m} \in \{0, 1\}^n$, Number of timesteps: $t$
>
> ---
>
> 1. Generate a binary mask $\mathbf{m} \in \{0, 1\}^n$.
> 2. Center the motif coordinates: $\mathbf{x_1} \leftarrow \mathbf{x_1} - \bar{\mathbf{x}}_1$
> 3. **for** $i = 0$ to $t - 1$ **do**
>    &emsp;1. *# diffusion procedure*
>    &emsp;2. *Overwrite the motif region: $\mathbf{x_t} \leftarrow \mathbf{x_t} \cdot \mathbf{m} + \mathbf{x_1} \cdot (1 - \mathbf{m})$*
>
>    &emsp;3. *# compute loss*
> 4. **end for**
> ---
>
> We will include this explanation in the revised manuscript to clarify the pipeline for motif integration.
>
>
>
> > **Q5:** Your current approach conditions enzyme generation on a single substrate molecule. Many practical enzymes, however, bind multiple substrates or cofactors simultaneously. How might you extend your current framework to handle multi-substrate scenarios or enzymes catalyzing multi-step reactions?
>
> **Response:** Thank you for the insightful comment. We agree that supporting multi-substrate reactions would broaden the model’s applicability. While our current design focuses on single substrate, our architecture is flexible and can be readily extended to multi-substrate scenarios.
>
> As a preliminary idea, we suggest using the same annotated motifs for each substrate, obtaining individual substrate-guided representations via EnzyAdapter, and aggregating them before passing them to the Transformer generator. This allows the model to capture multiple interaction contexts. A case study is shown below.
>
> **sequence:** LSPEEIEEIKANNQWAERTAALDKTVTLNPSLTLGDWTVDNTGGLDDPDAATRLCRGTIDLATGKIGSGGSVGEKDGGVTIGGLSLGVEEDGVLHGYLAEISASGATVRVPVRPDDTYRDLAARAQAQLGTSSDAATGATLTLTDIEVRNVGFIITASSA
>
> **substrate1 (binding affinity = -6.62):** COC(=O)c1c(OC/C=C/c2ccc(c(c2)c2onc(c2)C(=O)O)F)cccc1O
>
> **substrate2 (binding affinity = -6.4):** OCc1ccc(c(c1)N(c1ccnc(n1)Nc1cc(cc(c1)S(=O)(=O)C)N1CCOCC1)C)C
>
> We appreciate the suggestion and view flexible support for multi-substrate inputs as a valuable direction for future work.
>
>
>
> > **Q6:** The evaluation relies heavily on predicted functional metrics. How confident are you that these predicted metrics translate to real-world enzyme functionality? Have you considered evaluating or cross-validating your generated designs using orthogonal computational methods or literature-based validation?
>
> **Response:** Thank you for the thoughtful question. We agree that the reliability of predicted functional metrics is essential for evaluating the practical utility of generated enzymes. While our evaluation relies on predictive models, these metrics are grounded in experimentally validated methods:
>
> - kcat, EC Match Rate, and ESP Score are computed using deep learning models benchmarked against wet-lab data:
>   - CLEAN [2] (EC classification) has been validated on real enzymes like MJ1651 and SsFIA, and can predict novel EC numbers.
>   - UniKP [3] (kcat prediction) is trained on experimental kinetics data and has been validated with wet-lab results on tyrosine ammonia lyase (TAL).
>   - ESP score [6] quantifies enzyme–substrate interaction strength, with statistical testing for interpretability and confidence.
> - Binding affinity is computed using docking, which is physics-based, offering an independent, orthogonal validation of structural compatibility.
>
> In response to your suggestion, we also evaluated similarity between generated and native scaffolds using AAR and RMSD (see Q1). These additional metrics further support the plausibility of our results.
>
> We will include this discussion and the corresponding table in the revised manuscript, and thank you for encouraging a more comprehensive assessment.
>
>
>
> > **Q7:** The EnzyAdapter component and the two-stage training scheme are central to your method. However, similar adapter-based mechanisms and multi-stage fine-tuning procedures are increasingly common in generative modeling literature. Could you clarify explicitly the unique contributions or adaptations made to these elements specifically tailored for enzyme backbone generation?
>
> **Response:** Thank you for the insightful question. While we acknowledge that these elements are increasingly used in the generative modeling literature, our contribution lies in **how these components are adapted and integrated with enzyme-specific biological context**. We also contribute a suit of functional evaluation metrics and a benchmark. For a detailed explanation, please refer to our response to **Q3**.
>
>
>
> **Reference:**
>
> [1] Geffner T, Didi K, Zhang Z, et al. Proteina: Scaling flow-based protein structure generative models[J]. arXiv preprint arXiv:2503.00710, 2025.
>
> [2] Yu T, Cui H, Li J C, et al. Enzyme function prediction using contrastive learning[J]. Science, 2023, 379(6639): 1358-1363.
>
> [3] Yu H, Deng H, He J, et al. UniKP: a unified framework for the prediction of enzyme kinetic parameters[J]. Nature communications, 2023, 14(1): 8211.
>
> [4] Wang C, Qu Y, Peng Z, et al. Proteus: exploring protein structure generation for enhanced designability and efficiency[J]. bioRxiv, 2024: 2024.02. 10.579791.
>
> [5] Krishna R, Wang J, Ahern W, et al. Generalized biomolecular modeling and design with RoseTTAFold All-Atom[J]. Science, 2024, 384(6693): eadl2528.
>
> [6] Kroll A, Ranjan S, Engqvist M K M, et al. A general model to predict small molecule substrates of enzymes based on machine and deep learning[J]. Nature communications, 2023, 14(1): 2787.

---

> > ### Comment · Reviewer_F4KC · 2025-08-04
> >
> > Most of my concerns have been addressed, and I have accordingly raised my score.

---

> > > ### Author Response · Authors · 2025-08-05
> > >
> > > Thank you very much for your supportive and thoughtful response.
> > >
> > > We sincerely appreciate your encouraging feedback and are pleased that the additional experiments and evaluation metrics clarified the effectiveness of our enzyme backbone generation approach. We also found your suggestion on handling multi-substrate scenarios particularly insightful. Your comments on improving the clarity of our algorithmic description and the motivation behind the model design have significantly enhanced the readability of our manuscript.
> > >
> > > We are grateful for your constructive input and will carefully consider these directions in our future work.

---

### Official Review · Reviewer_GqqU · 2025-06-30

**Clarity:** 3
**Significance:** 3
**Originality:** 3
**Rating:** 4
**Confidence:** 5

**Summary:**

This paper proposes EnzyControl, a flow-matching method to design enzyme backbone structures conditioned on automatically mined functional sites and corresponding substrates. The proposed method shows good performance on many metrics, demonstrating its good performance in designing high-quality enzyme backbone structures.

**Questions:**

Please see above weaknesses.

**Ethical Concerns:**

["NO or VERY MINOR ethics concerns only"]

**Limitations:**

Please see above weaknesses.

**Quality:**

3

**Strengths And Weaknesses:**

Strengths:

1. The paper curates an enzyme-substrate complex benchmark EnzyBind from PDBbind, which is useful for either designing enzyme-substrate complexes  or substrate-binding enzyme design.

2. The proposed method shows good performance compared to strong baselines.

3. The writing of this paper is pretty good, making this paper easy to follow.

Weaknesses:

1. In line 47, the authors mentioned "existing benchmarks are mostly synthetic with limited experimental grounding and lacking evaluation protocols tailed to enzyme families". According to my best knowledge, EnzyBench is curated from PDB, which are all experimentally confirmed structures, and it is classified into fourth-level EC categories. So this statement is not right. However, I do think EnzyBind is meaningful as EnzyBench are composed of only enzymes while EnzyBind are enzyme-substrate complexes. The problem is in this paper, the authors didn't design enzyme-substrate complexes or designing substrate-binding enzyme backbones. Instead, they just model the substrate representation and then design an enzyme without modeling enzyme-substrate interactions. This way makes EnzyBind meaningless. As the way this paper used, EnzyBench can achieve the same goal.

2. In experiments, there are many metrics evaluated on the designed enzyme sequences. While this paper targets at designing enzyme backbone structures, they subsequently applied ProteinMPNN to generate an enzyme sequence to achieve evaluation. This is actually unfair to other baselines models that directly design enzyme sequences.

---

> ### Author Rebuttal · Authors · 2025-07-31
>
> Thank you for the extensive efforts on reviewing our submission and the valuable suggestion. Here we present a detailed response to address your concern.
>
> > **Q1:** In line 47, the authors mentioned "existing benchmarks are mostly synthetic with limited experimental grounding and lacking evaluation protocols tailed to enzyme families". According to my best knowledge, EnzyBench is curated from PDB, which are all experimentally confirmed structures, and it is classified into fourth-level EC categories. So this statement is not right.
>
> **Response:** We thank for your valuable feedback and clarify our intention and provide a more precise explanation below:
>
> - **Clarification on synthetic and predicted structures**: Our original statement aimed to highlight that many recent enzyme-related benchmarks—such as EnzymeFill [1] and ClipZyme [2]-include structures predicted by AlphaFold2 rather than experimentally resolved ones. These datasets, though useful, may not offer the level of structural accuracy needed for generative tasks that are highly sensitive to atomic detail.
> - **Specific limitation of EnzyBench for enzyme backbone generation**: While EnzyBench is indeed experimentally grounded and EC-classified, it provides only $C_\alpha$ coordinates, not full atomic backbone structures (e.g. $C_\beta$, $N$, $O$). This makes it less suitable for enzyme backbone generation tasks, which require complete and precise 3D geometries.
> - **Design considerations in EnzyBind**: To address these limitations, EnzyBind includes only experimentally determined structures. All entries are supported by literature references, and structures were resolved via X-ray crystallography or cryo-EM. We also applied a resolution filter during data curation to ensure structural quality.
>
>
>
> > **Q2:** The authors didn't design enzyme-substrate complexes or designing substrate-binding enzyme backbones. Instead, they just model the substrate representation and then design an enzyme without modeling enzyme-substrate interactions. This way makes EnzyBind meaningless. As the way this paper used, EnzyBench can achieve the same goal.
>
> **Response:** We thank the reviewer for the thoughtful comment. While EnzyBench provides a valuable evaluation protocol by filtering generated enzymes using substrate information post hoc, our method integrates enzyme-substrate interaction modeling directly into the generation process. Our response is as follows:
>
> - **Explicit interaction modeling during generation**: Our method integrates interaction signals directly into the generative process. Specifically, we jointly input the **substrate representation** and the **annotation motif**—which captures reaction-specific and functional context—into the model. These two sources of information are fused through the **EnzyAdapter** module, which facilitates **cross-modal information exchange** during backbone generation. This design allows the model to condition enzyme structure generation on both substrate characteristics and catalytic context. As a result, our task explicitly targets **substrate-binding enzyme backbone generation**, rather than generic scaffold generation.
> - **Interaction-aware evaluation**: To assess how well the generated enzymes bind the given substrates, we evaluated multiple interaction-specific metrics, including kcat, Binding affinity and ESP score. These metrics directly reflect the compatibility between enzyme and substrate.
> - **Empirical support via ablation studies**: Our ablation experiments (Table 4) demonstrate that removing either the **annotated motif** or the **EnzyAdapter** significantly reduces performance on the above interaction-related metrics. This confirms the importance of these components for learning enzyme-substrate interactions during generation.
>
>
>
> > **Q3:** In experiments, there are many metrics evaluated on the designed enzyme sequences. While this paper targets at designing enzyme backbone structures, they subsequently applied ProteinMPNN to generate an enzyme sequence to achieve evaluation. This is actually unfair to other baselines models that directly design enzyme sequences.
>
> **Response**: Thank you for the valuable comment. We apologize for the lack of clarity in the original manuscript. To ensure a fair comparison, **all baseline models were evaluated using the same pipeline**. We have revised the manuscript to make this evaluation strategy explicit. Below, we provide further clarification:
>
> - For all motif-scaffolding baselines, including those not originally designed for structure generation, we followed a unified evaluation protocol: the generated backbones were first passed through **inverse folding using ProteinMPNN**, then **all-atom structures were predicted with ESMFold**, and finally, all evaluation metrics were computed on the predicted structures.
> - For *EnzyGen*, which performs backbone–sequence co-design, we extracted the generated backbone structures and evaluated them using the **same pipeline**—inverse folding, followed by ESMFold-based structure prediction and metric computation—to maintain consistency across all methods.
>
> We again apologize for the earlier ambiguity and have updated the manuscript to clearly describe this standardized evaluation approach.
>
>
>
> **Reference:**
>
> [1] Hua C, Liu Y, Zhang D, et al. Enzymeflow: Generating reaction-specific enzyme catalytic pockets through flow matching and co-evolutionary dynamics[J]. arXiv preprint arXiv:2410.00327, 2024.
>
> [2] Mikhael P G, Chinn I, Barzilay R. Clipzyme: Reaction-conditioned virtual screening of enzymes[J]. arXiv preprint arXiv:2402.06748, 2024.

---

> > ### Comment · Reviewer_GqqU · 2025-08-05
> > **Response**
> >
> > First, I would like to thank the authors for clarifying my concerns.
> >
> > However, I still have the problem of how to use EnzyBind. I understand they used substrate representation during generation, but my point is they didn't directly model the enzyme-substrate complex, i.e. substrate-binding enzyme design as what their dataset is. For previous work like EnzyBench, it could also achieve this goal if we don't consider the complex design. From this aspect, the curation of EnzyBind seems meaningless.
> >
> > Therefore, I'd like to maintain my current score.

---

> ### Author Response · Authors · 2025-08-06
>
> Thank you for your thoughtful feedback. We truly appreciate the suggestion of a fully integrated pipeline that simultaneously generates ligand conformations, sequences, and structures. This is indeed a compelling and forward-looking direction. However, in this work, we pursue a **different yet complementary approach**, focusing specifically on **substrate-targeted enzyme backbone generation**. In our pipeline, we **generate sequences using ProteinMPNN**, and **obtain ligand conformations through docking-based methods**, which allows us to realistically model enzyme–substrate interactions without assuming prior knowledge of the binding pose.
>
> Our design philosophy emphasizes the incorporation of **biological priors** and **substrate-specific context** in a modular fashion, which brings both interpretability and flexibility. Our pipeline consists of the following key components:
>
> - **EC-Based Motif Annotation:** EC numbers capture catalytic function, which is fundamentally shaped by enzyme–substrate interactions. We group enzymes by EC number and extract conserved motifs within each cluster. These motifs are used as **functionally relevant structural anchors**, especially near catalytic centers, guiding the model to generate backbones that preserve essential functional regions.
> - **Substrate Conditioning via EnzyAdapter:** To effectively incorporate substrate-specific information, we introduce **EnzyAdapter**, a lightweight conditioning module. It contains a **cross-modal projector** to align substrate and enzyme embeddings and uses **cross-attention layers** to inject substrate-level guidance into the generation process. Importantly, this conditioning is achieved **without modifying the base structure generator**, making it easily extensible. This allows the model to generate enzyme backbones that are both structurally sound and tailored to the given substrate.
>
> Regarding **substrate 3D structures**, we deliberately chose **not to rely on known binding poses**. In practical enzyme design scenarios—especially when engineering novel functions—the binding conformation is typically unknown beforehand. Using an arbitrary substrate conformer during generation would still require docking to assess compatibility. For this reason, we compute **binding affinity post-generation via molecular docking**, which better reflects real-world design pipelines.
>
> Finally, we’d like to highlight a **technical distinction between our EnzyBind benchmark and EnzyBench**. EnzyBench provides only **$C_\alpha$ coordinates**, which limits its use in tasks requiring full 3D structural context. In contrast, EnzyBind includes **$C_\alpha$, $C_\beta$, $N$, and $O$ atoms**, enabling **SE(3)-equivariant modeling** and supporting more fine-grained structure-based enzyme generation. This richer atomic information is crucial for achieving high-quality backbone design, especially near functional sites.
>
> We sincerely appreciate your engagement and would be happy to further discuss any additional ideas or questions.

---

### Official Review · Reviewer_DtbU · 2025-07-02

**Clarity:** 3
**Significance:** 2
**Originality:** 3
**Rating:** 4
**Confidence:** 2

**Summary:**

This paper presents EnzyControl, a method for generating enzyme backbones conditioned on both functional catalytic motifs (extracted via MSA) and substrate molecules (represented as chemical graphs). The approach introduces EnzyAdapter, a modular component that injects substrate awareness using cross-attention, and employs a two-stage training strategy for stability and alignment. EnzyControl outperforms existing methods on structural and functional metrics, using a new benchmark dataset called EnzyBind curated from experimentally validated enzyme–substrate complexes.

**Questions:**

Have the authors tested the effect of motif misannotation?
Why is the substrate encoder (Uni-Mol) frozen?

**Ethical Concerns:**

["NO or VERY MINOR ethics concerns only"]

**Final Justification:**

The authors have addressed most of my concerns during the rebuttal period.

**Limitations:**

yes

**Quality:**

3

**Strengths And Weaknesses:**

Strengths: The method introduces a novel approach to enzyme design by incorporating substrate-aware generation using the modular EnzyAdapter architecture. The paper also provides a comprehensive evaluation, including ablation studies, comparison against strong baselines. The paper introduces EnzyBind, a curated dataset of 11k enzyme–substrate pairs, which will serve as a useful benchmark for future work.

Weaknesses: The method is currently limited to single-chain enzyme scaffolds, with no support for modeling multimeric or complex allosteric systems, which are common in many natural enzymes. This constraint reduces the method’s applicability to broader classes of enzymes.

---

> ### Author Rebuttal · Authors · 2025-07-31
>
> We sincerely appreciate the reviewers’ thoughtful comments and constructive suggestions. These insights have been invaluable in helping us improve the clarity, rigor, and scope of our manuscript. In response, we have carefully revised the paper and provide detailed point-by-point replies to each of the comments below.
>
> > **Q1:** The method is currently limited to single-chain enzyme scaffolds, with no support for modeling multimeric or complex allosteric systems, which are common in many natural enzymes. This constraint reduces the method’s applicability to broader classes of enzymes.
>
> **Response:** We appreciate your concern regarding the complexity of allosteric systems and multimeric enzymes. Our decision to focus on single-chain backbone design is based on the following considerations:
>
> - **Simplification and practicality**: From the substrate’s point of view, what matters is the immediate local environment it interacts with—namely, the binding pocket. This local region defines the key physicochemical interactions, regardless of whether it is formed by a single chain or assembled from multiple chains. Therefore, in our approach, the pocket is treated as a self-contained structural fragment, abstracted from the broader oligomeric context. Designing within a single-chain framework simplifies the problem while still capturing the essential features of the pocket environment. This makes the method more tractable and broadly applicable in practice.
> - **Data availability**: There is a significant imbalance in data: we identified only ~2,659 multimeric enzyme–substrate complexes with sufficient functional annotations in the PDB. In contrast, high-quality single-chain data is far more abundant, enabling more reliable training and evaluation.
> - **Extensibility of our approach**: In response to your suggestion, we explored an extension to multimeric assemblies via post-hoc binder design. Specifically, we first generate a single-chain enzyme using our method, and then apply RFDiffusion’s binder design module to create a complementary chain that binds to the enzyme surface. This results in a multimeric complex composed of the designed enzyme and a newly generated binding partner.
>
> We experimented with this pipeline and present preliminary results on multimeric complex formation below.
>
> **original backbone:** MELPKRRIRLLVLYTPEVEAGPLADPAKREAHIREVVAKVNELLKPFNIEIVLVDIISIGSNYDVDFSAPCEALRAQLEALVATKLKKEIDFDMAVVFGGESLAPCIEGFAALGADISTGRGVALAVLDPSDAEADARAVAAQILRLLGVTAPPERRVGPNGGDEDGVVWGEDGVEESLAWSLEQLRRYFEEHQPAEYLLPP
>
> **binder:** SGLERWKEIDENNQWEELTKELLAKQVYRPETNAATGATIIATGPAGAELGAALRAAYGPDPATLVGGVLPRPTTTGIGYAFLGGVQTPEELARIARLLVSDPTAAVAAYVMTAEDGRIHWDEAAGRAWLAE
>
>
>
> > **Q2:** Have the authors tested the effect of motif misannotation?
>
> **Response:** In ablation study (Table 4, w/o MSA), we have a similar experiment, where the motif (*i.e.*, MSA) is completely random. We can observe that this ablation shows sigificant performance drop.
>
> Following your instruction, we now include a new experiment where only 50% of motif residues are randomly perturbed. The results (shown below) demonstrate that (1) motif misannotation substantially affects model performance, and (2) our MSA-based annotation strategy is important for functional enzyme design.
>
> | perturbed rate | >0.5scTM | Designability | EC Match Rate | kcat   | Binding Affinity | ESP score |
> | -------------- | -------- | ------------- | ------------- | ------ | ---------------- | --------- |
> | 100%           | 0.8719   | 0.6863        | 0.4764        | 2.4615 | -6.4361          | 0.7183    |
> | 50%            | 0.8761   | 0.7023        | 0.4918        | 2.6540 | -6.6105          | 0.7238    |
> | 0%             | 0.8848   | 0.7160        | 0.5041        | 2.9168 | -6.9303          | 0.7334    |
>
>
>
> > **Q3:** Why is the substrate encoder (Uni-Mol) frozen?
>
> **Response:** Our decision to freeze the encoder was carefully considered based on several factors:
>
> - **Preserving generalization from large-scale pretraining.** Uni-Mol was pretrained on **209 million** molecular conformers, capturing extensive chemical structure and interaction knowledge. In contrast, our downstream dataset contains only about **11,000 enzyme–substrate pairs**, which is several orders of magnitude smaller. Fine-tuning such a large pretrained model on limited data would risk overfitting and degradation of its generalizable representations.
> - **Adaptation via a learnable projector.** To bridge the domain gap without modifying the frozen encoder, we introduce a learnable projector that maps Uni-Mol’s substrate embeddings into the enzyme representation space. This enables our model to incorporate substrate-specific information in a trainable and flexible way, while retaining the pretrained encoder’s robustness.
>
> We believe this approach offers a strong trade-off between leveraging large-scale chemical knowledge and adapting to the specificity of the enzyme generation task. We have clarified this rationale in the revised manuscript.

---

> > ### Comment · Reviewer_DtbU · 2025-08-06
> >
> > Thanks for your reply. Most of my concerns have been addressed, and I will maintain my assessment.

---

### Note · Authors · 2025-08-12

Dear Reviewers, ACs, SACs, and PCs,

We sincerely thank you for your time and constructive feedback. The review and rebuttal process has substantially strengthened our work. Below, we summarize the key contributions recognized by the reviewers, the clarifications and additional experiments provided during the rebuttal, and our planned revisions for the final version.

**Positive feedback from reviewers：**

- **Novelty & Significance** (DtbU, GqqU, F4KC, YKVy): The EnzyBind was recognized as a valuable resource for enzyme design. Our formulation of enzyme backbone generation conditioned on both motifs and substrates was noted as a meaningful and underexplored problem.
- **Technical contribution** (F4KC, DtbU): The integration of multiple architectures into a unified generative framework represents a clear advancement over prior enzyme design approaches.
- **Empirical evidence** (F4KC, DtbU, GqqU): The experiments were considered well-executed, informative, and strongly supportive of our claims.
- **Clarity & presentation** (GqqU, F4KC, YKVy, DtbU): The manuscript was praised for its clear exposition and thorough technical details.

**Key improvements during rebuttal：**

- **Clarifying novelty** (F4KC, YKVy, GqqU): Explicitly described how enzymology-specific insights are incorporated into the modeling pipeline.
- **Additional baselines** (F4KC, YKVy): Incorporated newer and more diverse baselines, demonstrating the superiority of our method.
- **Method extensions** (DtbU, F4KC): Outlined strategies for extending the approach to multi-substrate and multi-chain scenarios.
- **Effect of motif misannotation** (DtbU): Analyzed performance under varying rates of motif perturbation.
- **Technical details** (GqqU, F4KC): Added pseudo code and clarified the evaluation pipeline.
- **Evaluation reliability** (F4KC, YKVy): Included additional metrics (AAR, RMSD) and validated the reliability of predicted metrics.
- **Modeling interactions** (GqqU, YKVy): Clarified two key components for capturing enzyme–substrate interactions.

**Planned revisions for the final version：**

- **Additional results**: Include the impact of motif misannotation, results from additional baselines, and evaluations using two new metrics.
- **Expanded technical details**: Provide complete evaluation setup descriptions and pseudo code.
- **Clarified novelty and motivation**: Explicitly highlight the methodological novelty and articulate the underlying motivation in the method section.

---

### Decision · Program_Chairs · 2025-09-17

**Decision:**

Accept (poster)

**Comment:**

This paper proposes EnzyControl, a framework for enzyme backbone generation that integrates functional motif annotations from MSAs with substrate-specific conditioning through a lightweight EnzyAdapter module. The authors also contribute EnzyBind, a curated benchmark of over 11k experimentally validated enzyme–substrate complexes, and demonstrate improvements over baselines in structure quality, functional preservation, and catalytic metrics.

Strengths include the novelty of explicitly incorporating substrate information into backbone generation, the usefulness of the EnzyBind dataset, strong empirical evidence across multiple metrics, and generally clear exposition.

Weaknesses include reliance on predicted functional metrics without wet-lab validation, restriction to single-chain and single-substrate cases, modest architectural novelty relative to existing methods, and some debate about whether enzyme–substrate interactions are modeled explicitly enough. The most important reasons for acceptance are the strong technical execution, the integration of enzymology-specific priors into generative modeling, and the introduction of a high-quality benchmark that will likely support further research.

During the rebuttal period, the authors addressed concerns by adding new baselines (Proteina, Proteus, RFdiffusion-AA), providing experiments on motif misannotation, clarifying motif integration and evaluation pipelines, extending discussion of multi-substrate extensions, and offering docking-aware optimizations.  On balance, I recommend acceptance, as the paper makes a meaningful step forward in enzyme design and provides resources and methods that will be valuable to the community despite some limitations.